# In vivo cyclic overexpression of Yamanaka factors restricted to neurons reverses age-associated phenotypes and enhances memory performance

Alejandro Antón-Fernández [1,2,3] ✉, Marta Roldán-Lázaro[1,3], Laura Vallés-Saiz [1], Jesús Ávila [1,2] & Félix Hernández [1] ✉

In recent years, there has been success in partially reprogramming peripheral organ cells using cyclic Yamanaka transcription factor (YF) expression, resulting in the reversal of age-related pathologies. In the case of the brain, the effects of partial reprogramming are scarcely known, and only some of its effects have been observed through the widespread expression of YF. This study is the first to exclusively partially reprogram a specific subpopulation of neurons in the cerebral cortex of aged mice. The in vivo model demonstrate that YF expression in postmitotic neurons does not dedifferentiate them, and it avoids deleterious effects observed with YF expression in other cell types. Additionally, our study demonstrates that only cyclic, not continuous, expression of YF result in a noteworthy enhancement of cognitive function in adult mice. This enhancement is closely tied to increased neuronal activation in regions related to memory processes, reversed aging-related epigenetic markers and to increased plasticity, induced by the reorganization of the extracellular matrix. These findings support the therapeutic potential of targeted partial reprogramming of neurons in addressing age-associated phenotypes and neurodegenerative diseases correlated with aging.

There is ample evidence that even from late adulthood, the brain begins to develop age-related brain dysregulations[1]. During this complex process, the brain can undergo significant alterations in neuronal circuit function and plasticity, resulting in cognitive impairment and memory loss. Epigenetic changes in DNA, including alterations in DNA methylation patterns and histone modifications, have been implicated in both healthy ageing and age-related brain diseases, such as Alzheimer's disease[2–5].

For many years, ageing was considered an irreversible physiological process, but that view began to change in recent years. First, the studies by Yamanaka and Thomson succeeded in taking a differentiated adult cell back to a previous developmental stage[6–8]. Shinya Yamanaka, in particular, achieved in 2006 in reprogramming human fibroblasts to an embryonic-like stage by artificially inducing the expression of four embryonic exogenous transcription factors Oct4, Sox2, Klf4 and c-Myc (OSKM)[6], which were named Yamanaka factors (YF). Following this discovery, adult cells could be reprogrammed and dedifferentiated, reverting the phenotype of the cell to pluripotent stages up to what they called induced pluripotent stem cells (iPSCs)[7]. Although numerous techniques for in vitro iPSCs generation have been described in detail[9], it was not until 2013 that the first transgenic mouse could be subjected to in vivo cell reprogramming, as it was thought that the tissue microenvironment, conducive to cell differentiation, would, in principle, oppose reprogramming[10]. Using a Tet-ON-like system, a system with ubiquitous and conditional expression of YF that induced reprogramming in vivo was achieved. As expected, full reprogramming in vivo led to the generation of iPSCs and iPSC-derived teratomas[10]. To this end, in an attempt to avoid serious collateral problems, cyclic induction of YF in a transgenic mouse similar to that used previously was considered in 2016. The fact that reprogramming proceeds cyclically allows the induction of partial reprogramming without complete loss of cell identity by brief exposure to YF[11]. This partial reprogramming, in addition to not producing loss of cell identity, did not increase proliferation, but instead decreased DNA damage and cellular senescence, and even reversed epigenetic markers associated with physiological ageing[12].

[1]Centro de Biología Molecular Severo Ochoa (UAM-CSIC), Nicolás Cabrera, 1. Cantoblanco, 28049 Madrid, Spain. [2]Consejo Superior de Investigaciones Científicas (CSIC), Serrano 117, 28006 Madrid, Spain. [3]These authors contributed equally: Alejandro Antón-Fernández, Marta Roldán-Lázaro.
✉e-mail: aanton@cbm.csic.es; fhernandez@cbm.csic.es

This milestone opened a whole range of regenerative possibilities; therefore, a few years later, it was proposed to assess the effect in the Central Nervous System of YF ubiquitous overexpression in order to study potential rejuvenation effects on brain tissue. Thus, YF expression in mature mice partially prevented some ageing-associated changes. For example, it was found that cyclic expression of factors in adult mice from 6 months of age produced an increase in the synaptic plasticity of mature neurons in the dentate gyrus, which resulted in an improvement in long-term memory processes at 11 months of age[13]. Similarly, another study performed the same year, using the eye as a model tissue of the Central Nervous System, demonstrated that partial reprogramming by the expression of YF promotes regeneration in vivo[14].

To date, most of in vivo reprogramming studies have been performed in peripheral tissues such as hepatocytes and muscle[15,16]. However, as seen in previous studies, there are relevant differences between adult neurons (postmitotic cells) and other peripheral cells, as YF are not sufficient to induce direct reprogramming in order to achieve a pluripotent state from differentiated postnatal neurons[17]. This fact could be a potential advantage, as neuron reprogramming through only YF expression theoretically cannot dedifferentiate the cell, avoiding the potential formation of teratomas.

In order to harness this potential advantage and to verify it in vivo, we propose a study of a new transgenic mouse model with transcriptional activation of YF restricted to neurons under the postnatal α-CaMKII promoter. Alpha Calcium–calmodulin (CaM)-dependent protein kinase II is expressed in a specific subpopulation of excitatory neurons in different cerebral regions such as the hippocampus and neocortex, but not in other regions like cerebellum[18,19], which are highly involved in memory processing.

For the characterization of this model, an experimental group where the YF were induced during the first three months of life was used. Once the transgenic model was found to express YF transgene, we wanted to focus on older mice trying to assess potential in vivo reversion of age-associated brain features. The results showed for the first time the in vivo partial reprogramming of cortical neurons in a new transgenic mouse line. The induction of this process in adult animals for four months led the cerebral cortex to exhibit phenotypes associated with more youthful stages, subsequently inducing cognitive improvement.

## Results

### Characterization of CamKII-transgenic mice: neuron-restricted reprogramming in young mice

Since the transgenic model we used is entirely novel, we aimed to characterize it initially and verify that indeed Yamanaka factors (YF) were being effectively expressed. To do this, we began by studying the first offspring whose genotyping confirmed the presence of both transgenes. We maintained continuous activation of the doxycycline-inducible system from birth until the mice reached three months of age (when the nervous system should be fully formed), time when we collected samples.

To assure that exogenous YF were effectively expressed, quantitative PCR (qPCR) was performed against the sequence E2A-cMyc of the transgenes OSKM using hippocampus, neocortex and cerebellum samples. The results revealed a clearly higher expression in α-CaMKII-OSKM compared to transgenic control mice in hippocampus and neocortex, but no difference was observed in the cerebellum between transgenic and control mice, being almost undetectable (Supplementary Fig. 1a).

Attending the expression of one of the YFs (Klf4) in cerebral histological regions, we were able to determine more precisely the specific areas in which the promoters were active throughout the experiment (Supplementary Fig. 1b). It is important to note that due to the nature of the transgene, the expression of Klf4 (located in the third position within the gene order of the construct, after Oct4 and Sox2) implies that the other genes should have also been expressed in the neuron, so the expression pattern observed with Klf4 would correspond to that of the transgene as a whole.

The highest expression occurs in different neuronal layers of the hippocampal formation (DG, CA3, CA1, Subiculum), dorsal and ventral striatum, and neocortex (somatosensory, somatomotor, visual or orbital areas). Some expression was also observed in the thalamus but at a much lower level. No expression was found in other diencephalic regions, such as the globus pallidus, or in brainstem regions such as the substantia nigra, superior or inferior colliculus, cerebellum, or medulla. The varied pattern of α-CaMKII promoter expression throughout the different cerebral regions in these mice is in line with that described previously[20].

In order to better understand the molecular process behind, we have focused first on the hippocampal formation, which not only have exhibited one of the highest Klf4 protein expressions in this model but also because this region is involved in different key brain processes such as memory functioning or adult neurogenesis, both processes affected by ageing[21]. We obtained bulk transcriptomic data from the hippocampi of 8 different animals to which doxycycline was not administered since birth, allowing neuronal cells from double transgenic mice to express YF continuously since birth. RNA-seq analysis (ENA accession number is "PRJEB56610"), by using Deseq27[22] R software package, detected 1419 differentially expressed genes (DEG) ($q$ value < 0.05) in neuronal-reprogrammed animals regarding control mice. From all of them, 604 DEG resulted in decreased and 815 in increased expression. The MA-plot shows the log2 fold changes (M) between two conditions over the mean of normalised counts (A) for all samples (Supplementary Fig. 1c). In the heatmap (Supplementary Fig. 1d), is also shown how the expression of YF in a subpopulation of neurons, has led to significant changes in the transcriptome of a relevant group of genes. According to the Gene Ontology (GO) knowledgebase, which is the largest source of information on the functions of genes and proteins, and to the Kyoto Encyclopedia of Genes and Genomes (KEGG), we have been able to track down groups of differentially expressed genes (DEG) involved in certain cellular functions (Supplementary Fig. 1e-f; Supplementary Data 1). Data obtained revealed changes in the expression of genes related with regulation of nervous system development (46 DEG; 1.68E-07 p.adj), stem cell differentiation (21 DEG; 0.027 p.adj) or maintenance (17 DEG; 0.02 p.adj), central nervous system neuron differentiation (23 DEG; 1.74E-03 p.adj) or specifically in regulation of neuron differentiation (31 DEG; 1.91E-05 p.adj), would confirm that processes related with reprogramming in α-CaMKII-OSKM mice are taken place. In addition, transcription changes were found in genes related with: extracellular matrix organization (48 DEG: 5.09E-10 p.adj), structure (29 DEG; 1.29E-08 p.adj) or in the regulation of cell-cell adhesion (62 DEG; 2.86E-10 p.adj); in the regulation of synaptic organization (38 DEG; 1.68E-07 p.adj) and of synaptic structure or activity (39 DEG; 1.20E-07 p.adj); learning or memory (39 DEG; 4.30E-06 p.adj) and cognition (42 DEG; 3.85E-06 p.adj); dendrite morphology (26 DEG; 3.26E-05 p.adj) or development (34 DEG; 0.00019 p.adj), axogenesis (61 DEG; 3.07E-09 p.adj) as well as in regulation of neurogenesis (57 DEG; 6.93E-09 p.adj). It is important to note that all these functions are ultimately affected with ageing.

Furthermore, using a bioinformatics tool such as Ingenuity Pathway Analysis (IPA)[23], we were able to identify a common upstream regulator of downstream genes. IPA revealed several different types of upstream molecules (~400), including transcription regulators, transporters, cytokines, growth factors, kinases or various enzymes. Among all upstream regulators, HMG20A, IL4, FGFR, C1QA, KTMT2D, KAT2A or CREBBP are within the top 20 most significantly activated regulators ($P < 10^{-05}$ and Z-score ≥ 2; among the top 30 upstream regulator). These regulators have been associated with epigenetic modifications (KTMT2D, KAT2A), neuronal differentiation (HMG20A, FGFR, KTMT2D), neurodevelopment (C1QA, FGFR, CREBBP), the ageing process (IL4, FGFR) or memory functioning (KAT2A). On the other hand, APOE, TP63, Ptprd or PSEN1 were among the most significantly inhibited regulators ($P < 10^{-04}$ and Z-score ≥ 2; among the top 30 upstream regulator), and they are also involved in ageing or ageing-associated diseases (APOE, TP63, Ptprd or PSEN1), cell adhesion (Ptprd) or neural differentiation (PSEN1).

During these experiments the impact of continuous neuron-restricted reprogramming on mortality rate was analysed. Thus, in the case of these

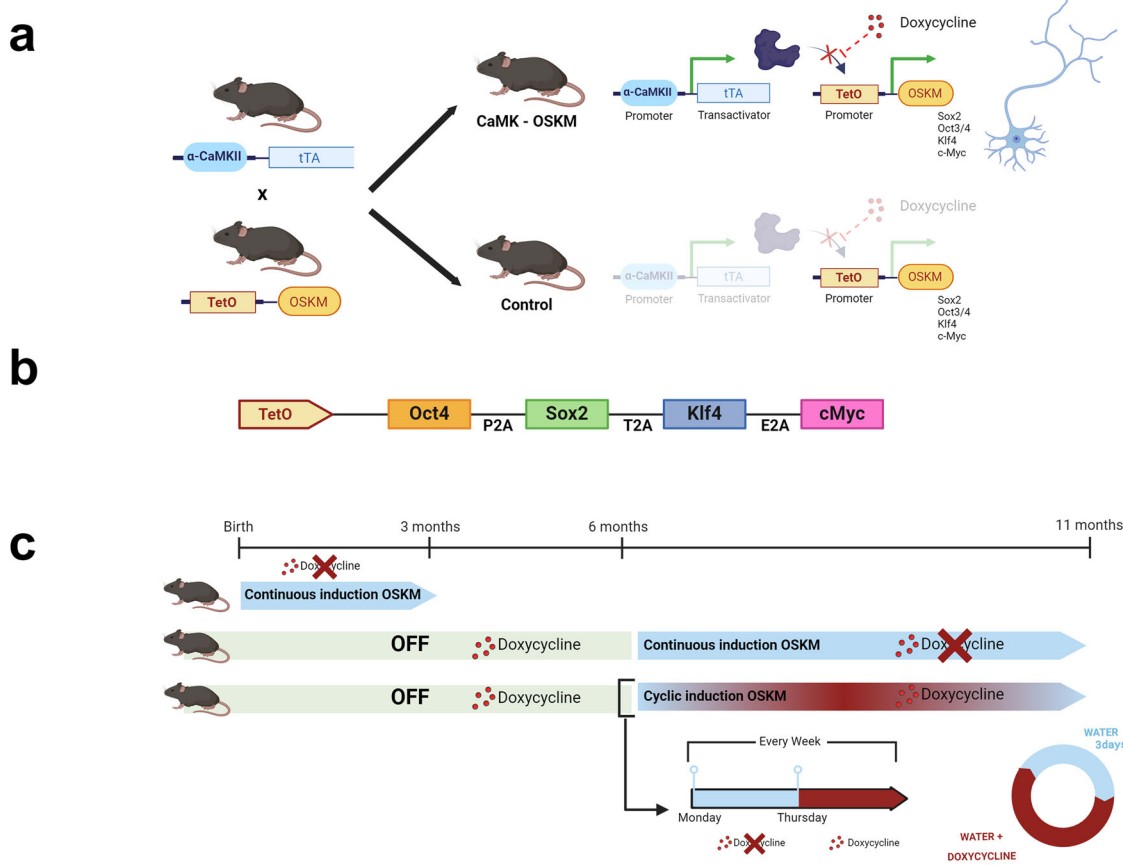

**Fig. 1 | New double transgenic mouse model for inducible expression of Yamanaka factors restricted to neurons expressing α-CaMKII promoter.**
**a** Crossbreeding was conducted between α-CaMKII-tTA and TetO-OSKM transgenic mice to generate the α-CaMKII-OSKM mice. **b** Schematic representation of the OSKM transgene showing the location of DNA sequences encoding 2 A peptides that separate each Yamanaka factor. **c** Temporal representation of the three OSKM treatments applied to the murine model in the present study. Doxycycline administration prevents binding between the transactivator tTA and the TetO promoter and thus inhibits transcription of Yamanaka factors.

young mice that underwent continuous induction of transgenes from birth, a high mortality rate (around 60%) was observed, along with the presence of hydrocephalus in some cases. However, for the mice in which the transgene system was only activated in adulthood at 6 months old, whose study will be described below, the mortality rate decreased to zero. Neither in the case of continuous induction nor in the case of cyclic induction did we find teratomas, confirming that the expression of YF in neurons did not lead to cellular dedifferentiation in vivo. This result is in line with the findings of Kim and colleagues[17].

**Neuron-restricted expression of Yamanaka factors in adult mice**
Next, we proceed to study the impact of neuron-restricted reprogramming on adult mice during long-term treatment, initiating the treatment in mice aged 6 months to nearly one-year-old, a protocol similar to that described previously[13]. Some mice were subjected to continuous factor expression by the continuous withdrawal of doxycycline over the 4-month treatment period, while another group of mice underwent cyclic doxycycline administration. This cyclic protocol involved administering water for 3 days per week and doxycycline for the remaining 4 days (Fig. 1). The cyclical expression of Yamanaka factors was confirmed by the histological study of Klf4 protein expression at different time intervals during a week cyclical protocol (Supplementary Fig. 2a-b). The analysis of immunofluorescence obtained from Klf4 Yamanaka factor expression have shown that indeed, after 3 days of induction, there is a significant increase (*p* value = 0.0043) in the expression of Klf4 Yamanaka factor (Supplementary Fig. 2c). This

induction returns to day 0 levels after 4 days of continuous doxycycline administration (*p* value = 0.0146).

**Characterization of the regional distribution of YF expression in adult mice**
To characterize the histological expression of YF in adult α-CaMKII mice, we employed antibodies targeting KLF4, just as in the previously shown study with young mice (Fig. 2a, b; Supplementary Fig. 3a). Quantitative analysis of immunohistofluorescence in adult α-CaMKII mice subjected to cyclic transgene induction from 6 to 11 months, demonstrated significantly higher YF expression compared to transgenic control mice (Fig. 2c, d). This difference was more pronounced in the deeper layers of the somatosensory neocortex (Fig. 2c). In the control group, YF expression was nearly absent in all regions of the cerebral cortex, as shown in Fig. 2a. Additionally, continuous transgene expression resulted in higher YF expression levels compared to transgenic control mice and adult α-CaMKII mice with cyclic YF expression. This difference was more evident in the hippocampal region (Supplementary Fig. 3b, c).

With respect to the regional distribution of Klf4 expression, adult α-CaMKII mice with cyclic and continuous expression of YF exhibited an immunofluorescence pattern of Klf4 protein expression similar to that of mice with continuous expression from birth at 3 months of age (Fig. 2b; Supplementary Fig. 3a). The highest expression was found throughout all the neocortex, mainly in deeper layers. Expression was also found in the hippocampus, subiculum, caudate putamen, piriform cortex and

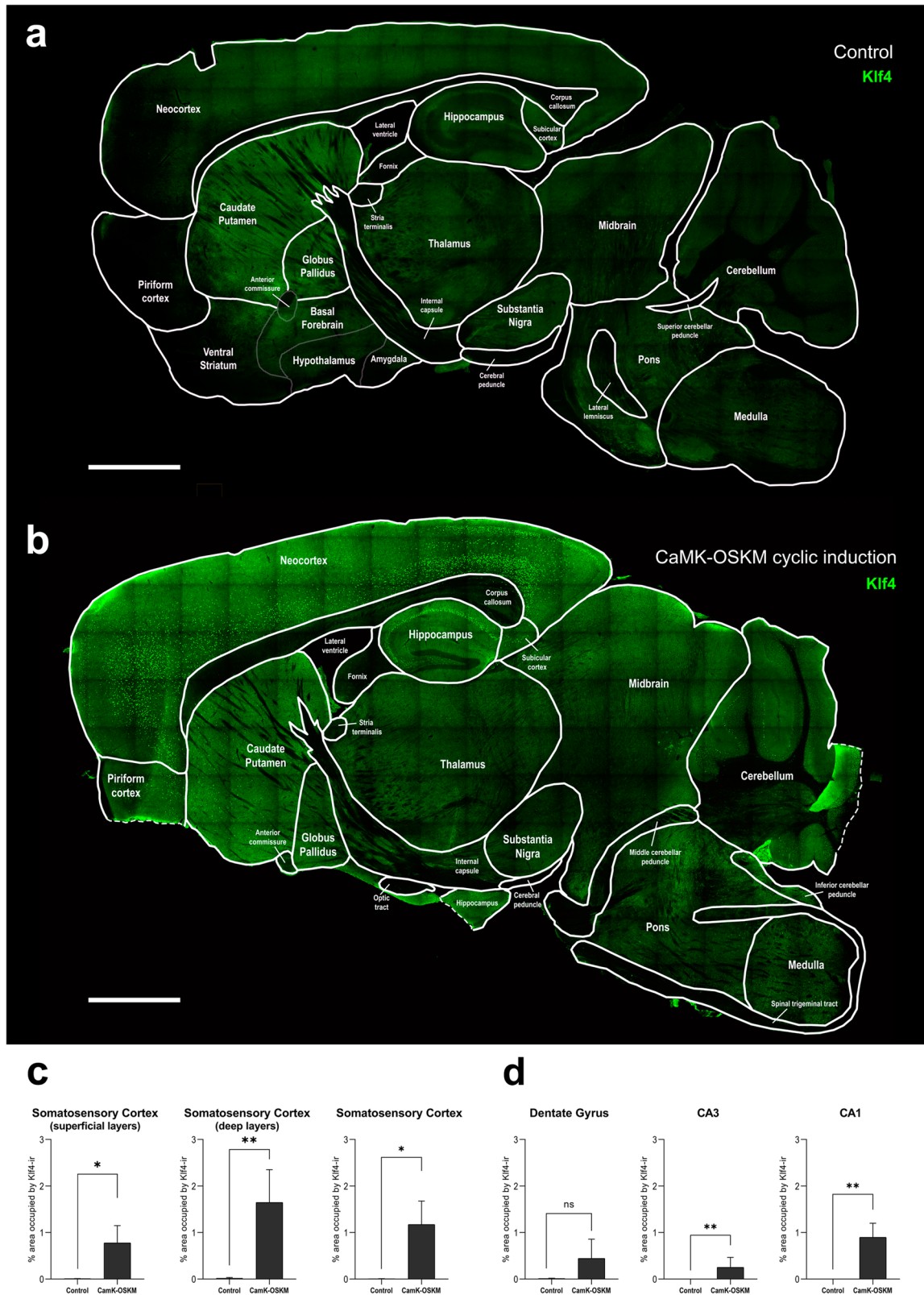

**Fig. 2 | Yamanaka factors expression in adult mice. a, b** Representative micro-photographs of the Klf4 Yamanaka Factor immunoreactivity (in green) obtained in sagittal brain sections from control 11 months old adult mice ($n = 5$) (**a**), α-CaMKII-OSKM mice with cyclic induction ($n = 5$) of transgene system from 6 to 11 months of age (**b**). Schematic outlines approximating the boundaries of some of the most relevant areas of the central nervous system have been overlaid on the microphotographs. Scale bar shown in a and b indicates 1000 μm. **c, d** Graphical representation of the mean ± SEM of the percentage of occupied area by Klf4-immunoreactivity in somatosensory neocortex, distinguishing deep layers from superficial (**c**) and different regions from hippocampal region, including CA1, CA3 and dentate gyrus (**d**). *$p < 0.05$ and ** $p < 0.01$ by Student's paired t-test.

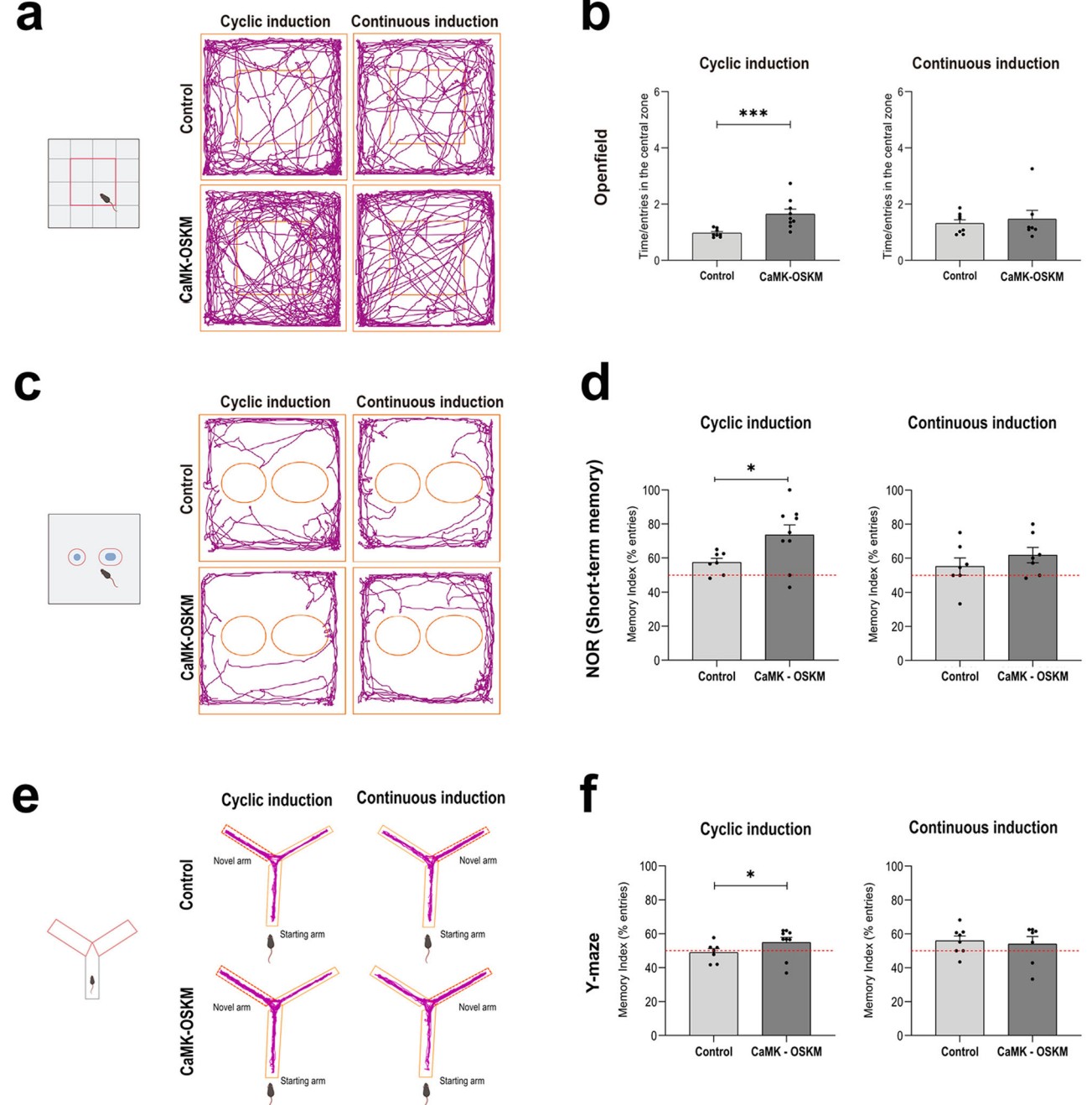

**Fig. 3 | Results of the behavioural test in neuronal partial reprogramming in adult mice. a, c, e** Schematic representation of the organisation of the space where the behavioural tests were performed, **a** Open field test, **b** Novel Object Recognition test and **c** Y maze test. Representative tracking maps obtained with Any Maze software, showing the trajectory of the centre of the rodent during the behavioural test. **b, d, f** Graphical representation of the mean ± SEM of the time (s) between the number of entries in the area analyzed (see Material & Methods) in different experimental groups. Control 11 months old adult mice ($n = 8$) and α-CaMKII-OSKM mice with cyclic induction ($n = 9$); control 11 months old adult mice ($n = 8$) and α-CaMKII-OSKM mice with continuous induction ($n = 7$). *$p < 0.05$, ** $p < 0.01$, *** $p < 0.001$.

thalamus, but cerebellum or other medullary nucleus seemed to lack YF expression.

### Behavioural characterization of YF expression in adult mice

We first analysed the animal's anxiety levels by observing their exploration behaviour within the central area of the box. Typically, rodents tend to remain close to the walls and avoid open spaces, a behaviour known as thigmotaxis[24]. As rodents age, it appears that they tend to spend less time in the central zone of the open field, which would translate to higher levels of anxiety[25,26]. The results showed statistically significant increase ($p$ value = 0.0003) in the time spent by the α-CaMKII-OSKM mouse in the central zone of the tray when they had cyclic treatment in comparison with the control group (Fig. 3a, b). As already described (see Materials and Methods), a short-term (2 hours) "novel object" recognition test was used to assess memory performance. The results of the test (Fig. 3c, d) showed a higher memory index in terms of time ($P$ value = 0,0149) and entries ($P$ value = 0,0204) in α-CaMKII-OSKM regarding the control group. Spatial memory was also evaluated through the Y-maze test (Fig. 3e, f), which showed improvements in the α-CaMKII-OSKM mice with cyclical administration of doxycycline with respect to the control group ($P$ value = 0,0422).

Therefore, cyclic activation of YF expression restricted to a subpopulation of neurons was enough to improve different types of memory in middle-age mice. Contrary to the cognitive effects found in α-CaMKII-OSKM adult mice with the cyclical induction of YF, continuous induction did not result in significant changes, either in the open field test or in the other tests conducted.

## Characterization of neuronal cyclic expression of Yamanaka factors in adult mice: transcriptomic study

Given that only the cyclical induction of YF in adult neurons yielded noteworthy improvements in cognition compared to continuous induction, we opted to focus on this approach, which would entail expression of YF within the α-CamKII promoter subpopulation of neurons.

We obtained transcriptomic data from both hippocampal and neocortical tissues of adult animals. Bulk RNA-seq analysis (ENA accession number is "PRJEB65922"), by using Deseq27[22] R software package, detected in total 94 differentially expressed genes (DEG) (q value < 0.05) in neuronal-reprogrammed animals regarding control mice (Supplementary Data 2). Out of all of them, the majority (~75%) showed decreased expression (70 DEG), while around 25% their expression was found to have increased. The gene expression data can be visualized in Fig. 4 where the MA-plot is shown for all samples in the neocortex and in the hippocampus (Fig. 4a, b; Supplementary Data 2). In the heatmaps (Fig. 4c, d), it is also shown how the expression of YF in a subpopulation of neurons, has led to significant changes in the transcriptome of a relevant group of genes.

According to the Gene Ontology (GO) knowledgebase, we have been able to track down groups of differentially expressed genes (DEG) involved in certain cellular functions (Fig. 4e, f; raw data in Supplementary Data 2). Firstly, it is important to note that, when examining the cellular component of ORA analysis (considering the both hippocampus and neocortex), the transcriptomic data revealed that reported changes were mainly located in specific neuron compartments, with the dendritic component standing out prominently (Fig. 4g). This finding confirms the specificity of YF expression solely in neurons. Data obtained revealed that genes with altered expression (qvalue < 0.05) were included in different biological processes, as regulation of nervous system development (Nectin3/Chrna4/Lrrtm4/Gabra5; 3.7E-04 padj) and neuron differentiation (Dab1/Trpc6/Brinp3/Neurog2; 5.5E-04 padj) that indicate functions compatible with a reprogramming process. Moreover, in general many of these differentially expressed genes (DEG), could be primarily grouped into alterations in processes related to the extracellular matrix (e.g. ECM organization Itga8/Col15a1/Adamts16/Fbln2/Grem1/Col22a1; 5.45E-04 Padj) and cell adhesion (e.g. regulation of cell substrate-adhesion; Col26a1/Pcsk5/Thy1/Ajap1/Fbln2/Ppm1f/Grem1, 7.27E-06), neuronal activity involving different classes of neurotransmitters (13 different processes, e.g., neurotransmitter receptor activity; Chrna4/Chrnb3/Gabra5/Htr2a/Hrh3, 2.96E-05 padj), cognition (Itga8/Chrna4/Gabra5/Htr2a/Hrh3/Jun, 9.4E-04), and processes always associated with neuronal structures and functions, with a particular emphasis on post-synaptic processes (e.g, postsynaptic specialization; Nectin3/Itga8/Chrna4/Dab1/Lrrtm4/Gabra5/Als2/Lzts3, 9.53E-05 padj).

Taking into account the hippocampus and the neocortex separately, the expression of these factors seems to have led to a somewhat more intense reprogramming process in the second region compared to the hippocampal region, considering the number of genes with altered expression (43 DEGs in the hippocampus vs. 59 DEGs in the neocortex). Among the genes with the most statistically significant differential expression, notable examples relate to the extracellular matrix, with collagen alpha 1 type XXVI (Col26a1; 7.5831E-11 padj) in the hippocampus and collagen alpha 1 type XXII (Col22a1; 1.63E-4 padj) in the neocortex. Furthermore, zinc finger proteins, like Zfp804b (3.6816E-06 padj) in the neocortex and Zfp386 (4.94E-09 padj) in the hippocampus, are noteworthy. Given their capacity to bind to chromatin, these proteins are believed to play a central role in neuronal reprogramming processes[27] and the latter have been involved in silencing LINE-1 elements[28,29]. In addition, there have been genes whose expression has been found to be altered in both cortical areas, such as Glis3 (another

zinc finger protein) and Fbln2. The expression of the Glis3 gene, functionally involved in reprogramming processes[30] and known to increase with aging[31,32], was significantly reduced in both hippocampal and neocortical regions in this study. Fbln2 is an extracellular matrix protein with roles in tissue remodelling and embryonic development[33,34].

## Impact on neuronal activity related with memory tasks

Since transcriptomic studies have revealed significant effects of partial reprogramming on regulatory neuronal activity genes we aimed to investigate, in a more specific manner, how these changes in neuronal activity may have contributed to the cognitive improvement observed in these animals during partial reprogramming. For this purpose, we conducted histological analyses on tissue samples using an immediate early gene c-Fos marker. In neurons, c-Fos expression is induced under conditions of neuronal plasticity, including learning and memory[35]. It has been widely used as a neuronal activity marker since they are rapidly and transiently induced by neuronal stimuli in the brain[36]. In this study, the mice were immediately perfused upon completion of the memory test. In this way, we were able to study the activity levels of the memory circuits during the execution of these memory tests. The analysis of c-Fos-immunoreactive cells was focused on the hippocampus (Fig. 5a), due to its essential role on recognition/spatial memory performance. We found a higher number of neurons active (c-Fos-immunoreactive) just after memory test performance in α-CaMKII-OSKM transgenic mice regarding transgenic control mice, in both granular cell layer of dentate gyrus and in the pyramidal cell layer of CA1 (Fig. 5b). This result potentially indicates more active circuits during memory testing due to YF expression restricted to neurons.

Additionally, we have found in these mice a significant inverse correlation ($R = 0.964$) between levels of Klf4 expression and density of c-Fos-immunoreactive cells (Supplementary Fig. 4). Excessive expression of Klf4 led to lower increase of c-Fos-immunoreactive cell density in the hippocampus of α-CaMKII-OSKM transgenic mice. These results are consistent with those found in α-CaMKII-OSKM mice with continuous induction of the YF. In these mice, where Klf4-YF expression is continuous, we observed worse cognitive performance compared to those with cyclical induction. All these results underscore once again the importance of the level of induction of the YF. Moderate rather than excessive induction is what achieves beneficial effects on the cognition of aged mice.

## Extracellular matrix reorganisation

Considering significant changes previously identified in the extracellular matrix (ECM) as a result of reprogramming in young α-CaMKII-OSKM mice (Supplementary Fig. 1, Supplementary Fig. 5), which led to an overall reduction in its structure, in addition to transcriptomic data obtained from α-CaMKII-OSKM adult mice showing significant alterations in genes related to ECM, we aimed to investigate whether partial reprogramming in adult mice would result in youthful ECM reorganization. Thus, we studied the expression of the cartilage-specific core protein proteoglycan (aggrecan), which binds to specific proteoglycans and allows visualisation of the so-called perineuronal extracellular matrix networks (Fig. 6a and d from the overall panoramic view). Immunoreactivity analysis for this protein was carried out in both the neocortex and hippocampal formation at both experimental groups (control and α-CamK-OSKM adult mice with cyclic overexpression of YF). Figure 6a shows representative microphotographs of the neocortex using an antibody against aggrecan protein, where, to facilitate the analysis, the supragranular layers (layers I-IV) have been distinguished from the infragranular layers (layers V and VI). The results have shown a prominent inclination towards an overall reduction in the percentage of area occupied by the aggrecan protein across the entire neocortex ($p$ value = 0.0518), attaining statistical significance within the deeper neocortical layers (V and VI) among α-CaMKII-OSKM mice in comparison to the control group ($p$ value = 0.0258; Fig. 6b).

In line with these results, the analysis of the density of perineuronal net units also shows a significant reduction following the induction of partial reprogramming in α-CaMKII-OSKM mice (Fig. 6c). In contrast to the

**Fig. 4 | Results of RNAseq performed in α-CaMKII-OSKM adult mice with cyclical induction of YF expression. a**, **b** MA-plot from neocortex (**a**) and hippocampus (**b**) samples which represent genes coloured in blue that have *q* values less than 0.05. Points which fall out of the window are plotted as open triangles pointing either up or down. Heatmap from neocortex (**c**) and hippocampus (**d**) samples. Data are displayed in a grid where each row corresponds to a gene and each column to a sample (from two different conditions). The colour and intensity in the heatmap represent changes of gene expression from the list of genes with *q* value in the Principal Component Analysis (PCA). Emapplot from neocortex (**e**) and hippocampal (**f**) samples, showing an enrichment Map for enrichment result of over-representation test or gene set enrichment analysis. **g** GO term analysis (cellular component) of altered genes.

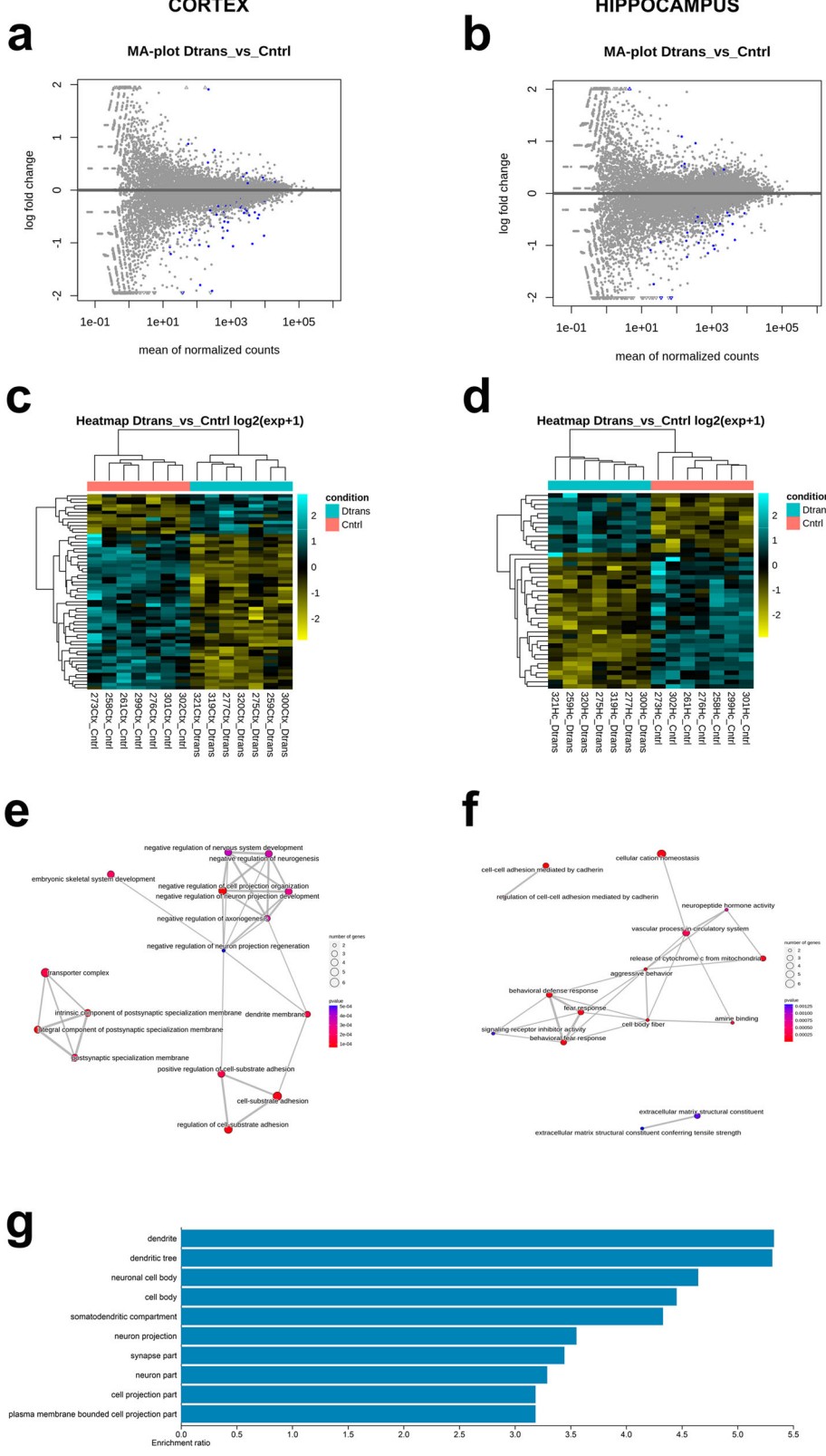

neocortical areas, noteworthy statistical differences were not found in the hippocampus between both experimental groups for the area occupied by the immunoreactive aggrecan matrix (Fig. 6e), nor regarding the density of PNN units (Fig. 6f). This general reduction found by immunofluorescence detection in aggrecan-immunoreactive extracellular matrix has been corroborated by Western blot technique (Supplementary Fig. 6a, b). We observed a highly significant decrease ($P = 0.0001$) in its expression following the cyclical induction of the Yamanaka factors. These data confirm the significant role of extracellular matrix reorganization during cyclical reprogramming processes in the brain.

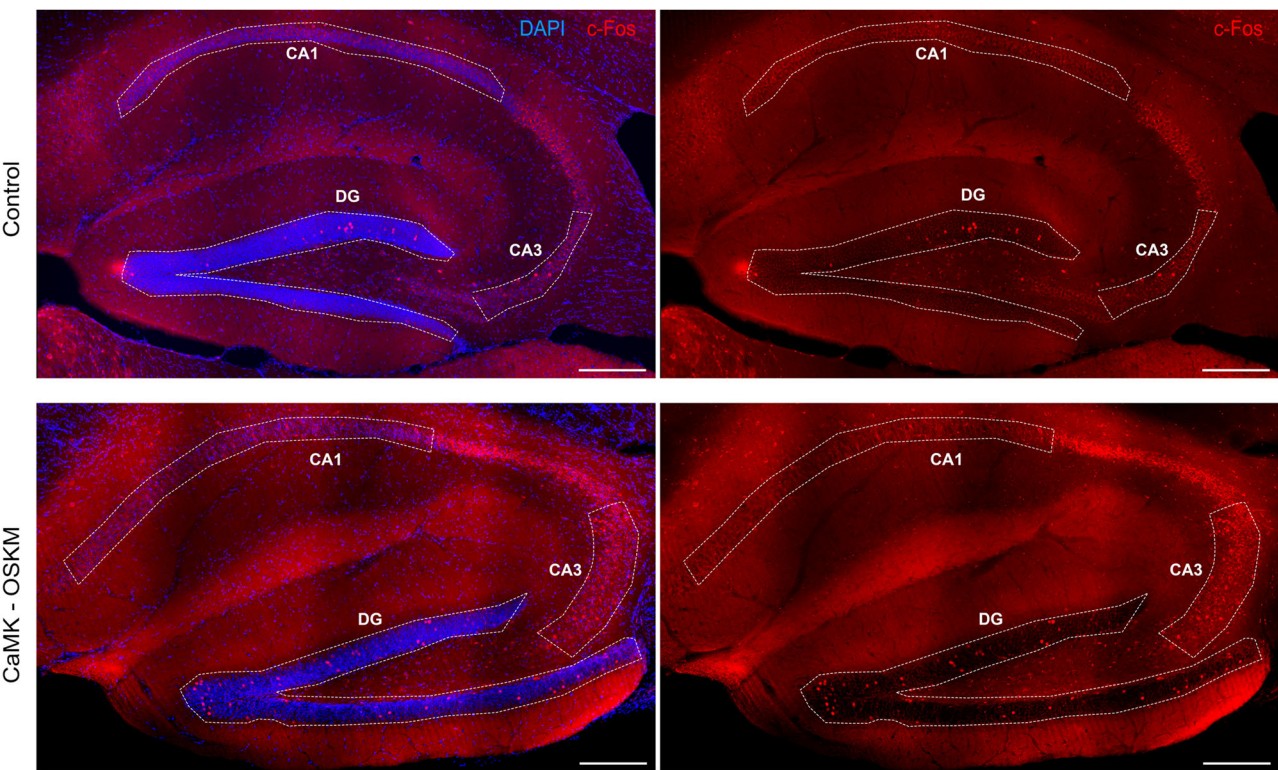

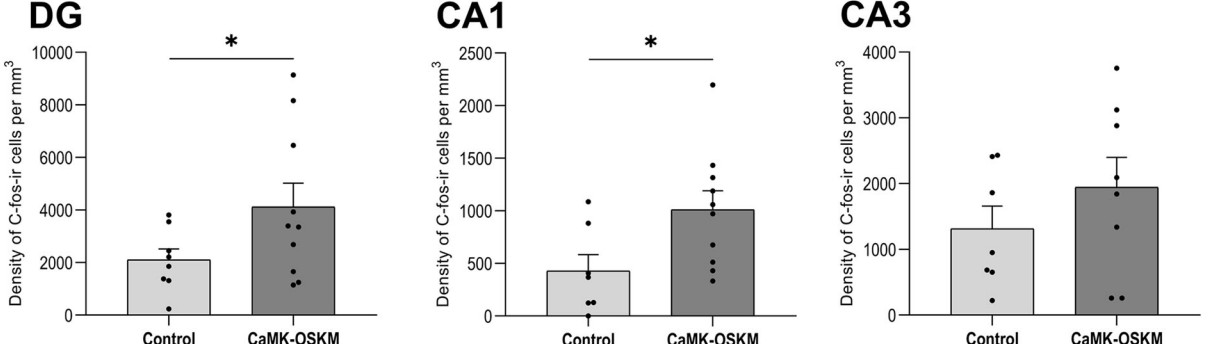

**Fig. 5 | Immunofluorescence results for c-Fos expression in cyclic neuronal partial reprogramming in adult mice. a** Representative microphotographs of immunoreactivity obtained for c-Fos protein expression (in red) in the hippo-campus region from control ($n = 8$) and α-CaMKII-OSKM mice ($n = 10$). Scale bar indicates 200 μm. **b** Graphical representation of the mean ± SEM of c-Fos-immunoreactive cells per mm³ at different neuronal layers of the hippocampal region (Dentate gyrus, DG; CA1; CA3). *$p < 0.05$.

## Impact of in vivo partial reprogramming in neuronal maturation and adult neurogenesis

In vitro studies have demonstrated that YF alone is not sufficient to induce neuronal dedifferentiation[17]. In this study, we decided to try to validate these findings in vivo, determining whether, under YF expression, neurons could undergo dedifferentiation. For this analysis, we employed a distinct set of antibodies. Doublecortin protein (Dcx) is expressed in migrating neuro-blasts and immature neurons, making it a reliable marker for adult neu-rogenesis. In general, in adult rodents, it is only possible to find immature neurons in regions where neurogenesis occurs, which are typically only two, one of which is the subgranular zone (SGZ) of the hippocampal dentate gyrus. Thus, we tried to detect doublecortin labelling outside of the sub-granular zone in α-CaMKII-OSKM mice. This would indicate that pro-cesses of dedifferentiation owing to YF expression from mature neurons to a previous state of maturation have occurred. In these animals, throughout the cerebral cortex we only observed doublecortin labelling in the sub-granular zone, similar to what was seen in the control group. We did not find these cells in the rest of the hippocampus or the neocortex. Moreover, we

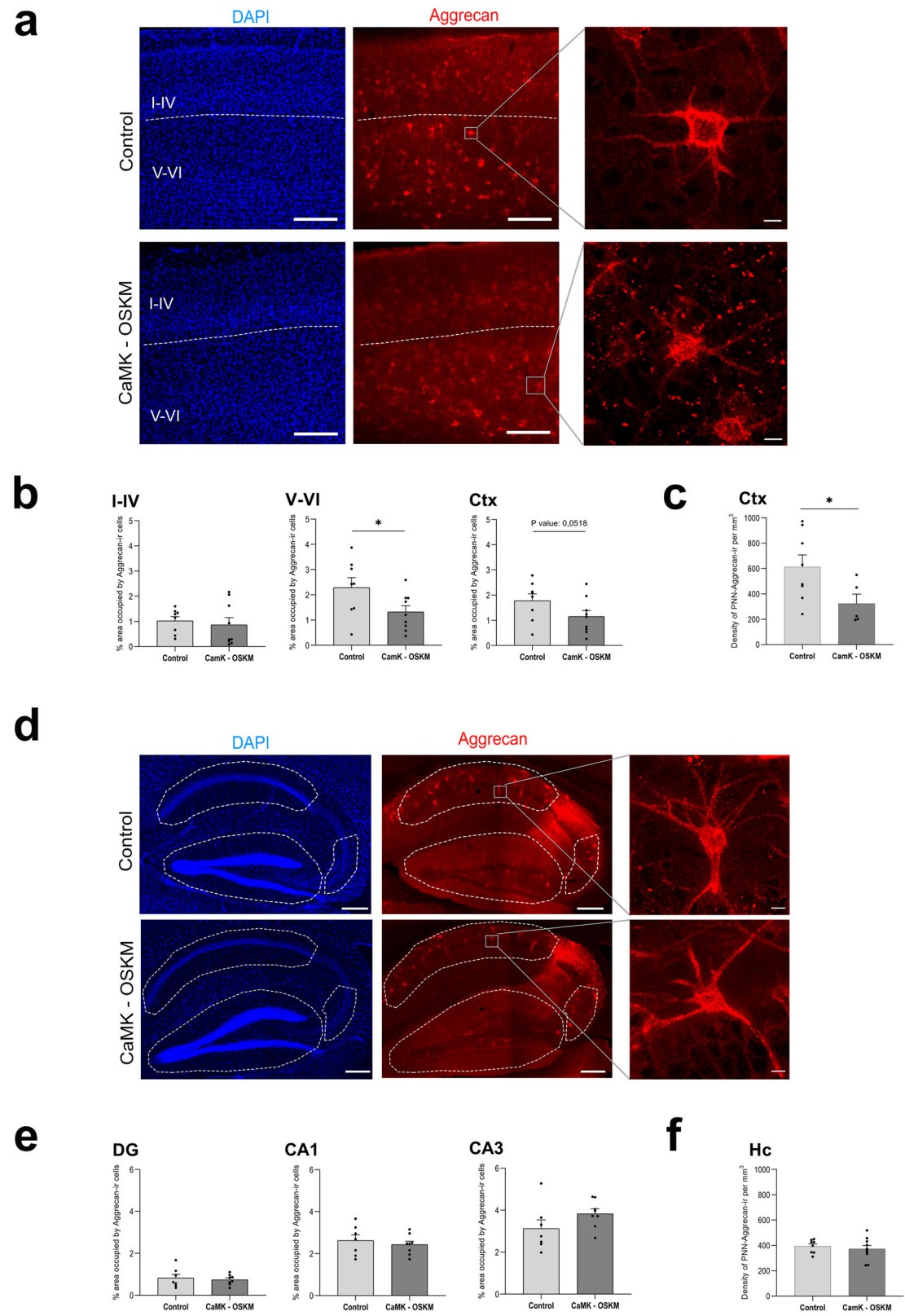

observed no differences in the density of Dcx-immunoreactive cells between the control and α-CamKII-OSKM mice in SGZ (Supplementary Fig. 7a, d). Furthermore, we aimed to study other markers of earlier neuronal development such as the intermediate progenitor marker T-box brain gene 2 (Tbr2) to ensure that partial reprogramming was not regressing to even earlier stages than those identified by the doublecortin marker. The results in

the SGZ revealed no differences in Tbr2 expression between the transgenic and control groups (Supplementary Fig. 7b, e). Additionally, 5-chloro-2′-deoxyuridine (CldU) was administered three weeks before perfusion for each mouse in both experimental groups to identify cells that were newly generated at that time. Results have shown how the number of three weeks-old cells labelled with CldU was not significantly changed in the DG of the α-

**Fig. 6 | Immunofluorescence results for Aggrecan expression in cyclic neuronal partial reprogramming in adult mice. a** Representative microphotographs of immunoreactivity obtained for Aggrecan protein expression (in red) in the somatosensorial neocortex from control ($n = 8$) and α-CaMKII-OSKM mice ($n = 9$). On the right side of the panel are enlargements of the panoramic view displaying the structure of perineuronal networks formed by the extracellular matrix. Scale bar shown in A indicates 200 μm and 15 μm in the magnification. **b** Graphical representation of the mean ± SEM of the percentage of area occupied by Aggrecan-immunoreactive cells per mm³ in total volume of the somatosensory neocortical region and at different cortical layers. *$p < 0.05$. **c** Density of perineuronal net units (PNNs) in the somatosensorial neocortex (number of Aggrecan-immunoreactive PNNs in each brain slice by the volume of the somatosensory neocortical region, *$p < 0.05$). **d** Representative microphotographs of immunoreactivity obtained for Aggrecan protein expression (in red) in the hippocampus of the different murine models. On the right side of the panel are enlargements of the panoramic view displaying the structure of perineuronal networks formed by the extracellular matrix. Scale bar shown in (**a**) indicates 200 μm and 15 μm in the magnification. **e** Representation of the mean ± SEM of the percentage of area occupied by Aggrecan immunoreactive per mm³ at different neuronal layers of the hippocampal region (Dentate gyrus, DG; CA1; CA3). **f** Density of perineuronal net units (PNNs) in total hippocampus (number of Aggrecan-immunoreactive PNNs in each brain slice by the volume of the area analysed).

CaMKII-OSKM, indicating that partial reprogramming by YF not only does it not appear to influence adult neurogenesis itself, but it also would not affect the proliferation of new cells in adult mice (Supplementary Fig. 7c). Differences in the density of CldU-labelled cells between control and α-CaMKII-OSKM adult mice in the somatosensory neocortex were not found either (Supplementary Fig. 7c, f).

### Epigenetic changes

Considering that the reprogramming induced by Yamanaka factors correlates with epigenetic changes[7,37], in this study we aimed to verify whether the selective partial reprogramming of neurons led them to more youthful epigenetic states. Early studies in rats showed that methylation of histones H3 and H4 changes gradually with increasing age[38]. Moreover, a systematic study of posttranslational modifications of histones in the brain of senescence-accelerated prone mouse 8 (SAMP8) model revealed a significant decrease of H4K20me3 marker during ageing[39]. Here, we have found a remethylation of this epigenetic marker at H4 after cyclic neuronal induction of YF expression in adult mice (Fig. 7). The results showed that H4K20me3 (histone 4 lysine 20 trimethylation) marker increases in α-CamKII-OSKM adult mice overexpressing YF in a cyclic manner regarding control mice throughout all the neocortex layers ($p$ value = 0.0124; Fig. 7a, b), as well as in CA1 ($p$ value = 0.0372) and CA3 ($p$ value = 0.0288) pyramidal cell layers at the hippocampal region (Fig.7c, d), but not in the granular cell layer of DG. According to previous studies these epigenetic changes in areas where partial reprogramming is taking out would lead to a more youthful epigenetic pattern in those reprogrammed cortical neurons[39].

### Discussion

In vivo ubiquitous expression of Yamanaka Factors (YF) has been achieved through partial cellular reprogramming reverting some age-associated phenotypes in different tissues[12,40], including the brain[13]. However, the ubiquitous expression, mainly due to its continuous induction, leads to premature death and the occurrence of teratomas[10]. To study, for the first time, the unique in vivo partial reprogramming of a specific neuronal subpopulation in mice, we developed a novel transgenic model with inducible expression of YF but restricted only to a large neuronal population (α-CaMKII neurons). This new reprogramming strategy applied in the present study to middle-age mice has enabled the cortex to attain a more youthful phenotype, additionally achieving significant cognitive enhancement in various behavioural and memory tests. These achievements have been reached without issues of survival problems, increased neuronal proliferation or teratoma formation. Thus, our results validate for the first time in an in vivo model, the findings obtained in vitro in the study conducted by Kim et al.[17]. They described how the expression of YK alone is insufficient to dedifferentiate neurons. This process, unlike the rest of the body´s cells, requires the induction of cell proliferation by supressing molecules such as p53.

To characterise this new transgenic mouse model, we have initially continuously activated the doxycycline-inducible system from birth to 3 months of age, when the mouse would already have a fully developed and mature brain. RNA sequence analysis confirmed that exogenous YF were effectively expressed resulting in changes in the expression of genes related to the regulation of nervous system development, stem cell differentiation or maintenance, specifically in the regulation of neuronal differentiation. These changes confirmed that partial reprogramming processes must be occurring in α-CaMKII-OSKM mice. In addition, differences detected in some upstream regulators that have been involved in epigenetic modifications like Histone-lysine N-methyltransferase 2D (KTMT2D), shows that in the process of partial reprogramming, changes must be taking place at the epigenetic level. In any case, one of the elements most significantly altered according to the transcriptomic results has been those related to the extracellular matrix structure. In fact, histological studies showed a widespread reduction of the extracellular matrix in the neocortex and hippocampal regions (see Supplementary Fig. 5). These changes reflect a reorganization of the extracellular matrix, likely shifting it toward patterns more characteristic of younger stages, which is characterized by reduced expression of its components[41,42], as is the case here.

Moreover, the continuous induction of YF from birth to 3 months of age appeared to have a deleterious effect on animal survival, likely due to neurodevelopmental issues stemming from incomplete maturation of the nervous system. However, the continuous or cyclic induction of the transgene system from 6 months, when the mouse has already reached adulthood, until 11 months of age, did not result in any increase in mortality or teratoma formation. The fact that ectopic expression of the YF in postmitotic cells such as neurons is insufficient to bring them to an embryo-like stage[17], may explain the absence of teratoma that typically arise when YF are expressed ubiquitously in an in vivo model[10]. Furthermore, we can rule out the possibility of self-inhibition of the reprogramming process due to the cell identity promoter (in this case, CamKIIα), as we have showed a significant expression of the YK in both the cyclic and, particularly, the continuous induction, indicating that the system is not inhibited, and the factors are successfully expressed.

Once verified that both cyclic and continuous expression of YF in adult α-CaMKII-OSKM mice did not cause problems in terms of mortality or teratoma formation, we conducted a battery of behavioural tests on the mice. Memory processing is significantly affected by ageing[43], even from middle age[1,44,45], thus we assess memory performance by carrying out tests such as the novel object recognition test or spatial memory test. The results showed that continuous induction of YF did not lead to cognitive improvement, whereas partial reprogramming, which involved cyclic expression of YF, resulted in a significant improvement of performance in both object recognition and spatial memory tests. Partial reprogramming of a restricted population of cortical neurons has thus been sufficient to induce a significant cognitive improvement in transgenic mice. As in previous studies[12], cyclic induction of YF seems to be more effective in obtaining reversal of age-associated phenotypes than continuous YF expression. It is also important to consider that, given the significant cellular stress involved in the reprogramming process[46], its continuous activation over extended periods may counteract the cognitive benefits that cyclic activation produces.

Taking these results into account, we focused on employing the cyclic induction strategy to better understand the processes underlying cognitive enhancement and evaluate the potential of neuronal partial reprogramming to restore youthful brain function and structure. The transcriptomic results

**Fig. 7 | Immunofluorescence results for H4K20me3 expression in cyclic neuronal partial reprogramming in adult mice. a** Representative microphotographs of immunoreactivity obtained for H4K20me3 expression marker (in green) in the somatosensory neocortex from control ($n = 8$) and α-CaMKII-OSKM mice ($n = 9$). On the right side of the panel, enlargements are displayed, showing H4K20me3-immunoreactivity expression in a group of cells present in the deeper neocortical layers. Scale bar shown indicates 200 μm and 15 μm in the magnification. **b** Representative percentage of mean intensity (arbitrary units) obtained from H4K20me3 immunoreactivity in neocortex, distinguishing supragranular layer (I-IV) and infragranular layers (V-VI). *$p < 0.05$. **c** Representative microphotographs of immunoreactivity obtained for H4K20me3 expression marker (in green) in the hippocampal region of the different murine models. On the right side of the panel, enlargements are displayed, showing H4K20me3-immunoreactivity expression in a group of cells present in the pyramidal neuronal layer of the CA1 region. Scale bar shown indicates 200 μm and 15 μm in the magnification. **d** Percentage of mean intensity (arbitrary units) obtained from H4K20me3 immunoreactivity in hippocampus, distinguishing CA1, CA3 and dentate gyrus (DG). *$p < 0.05$.

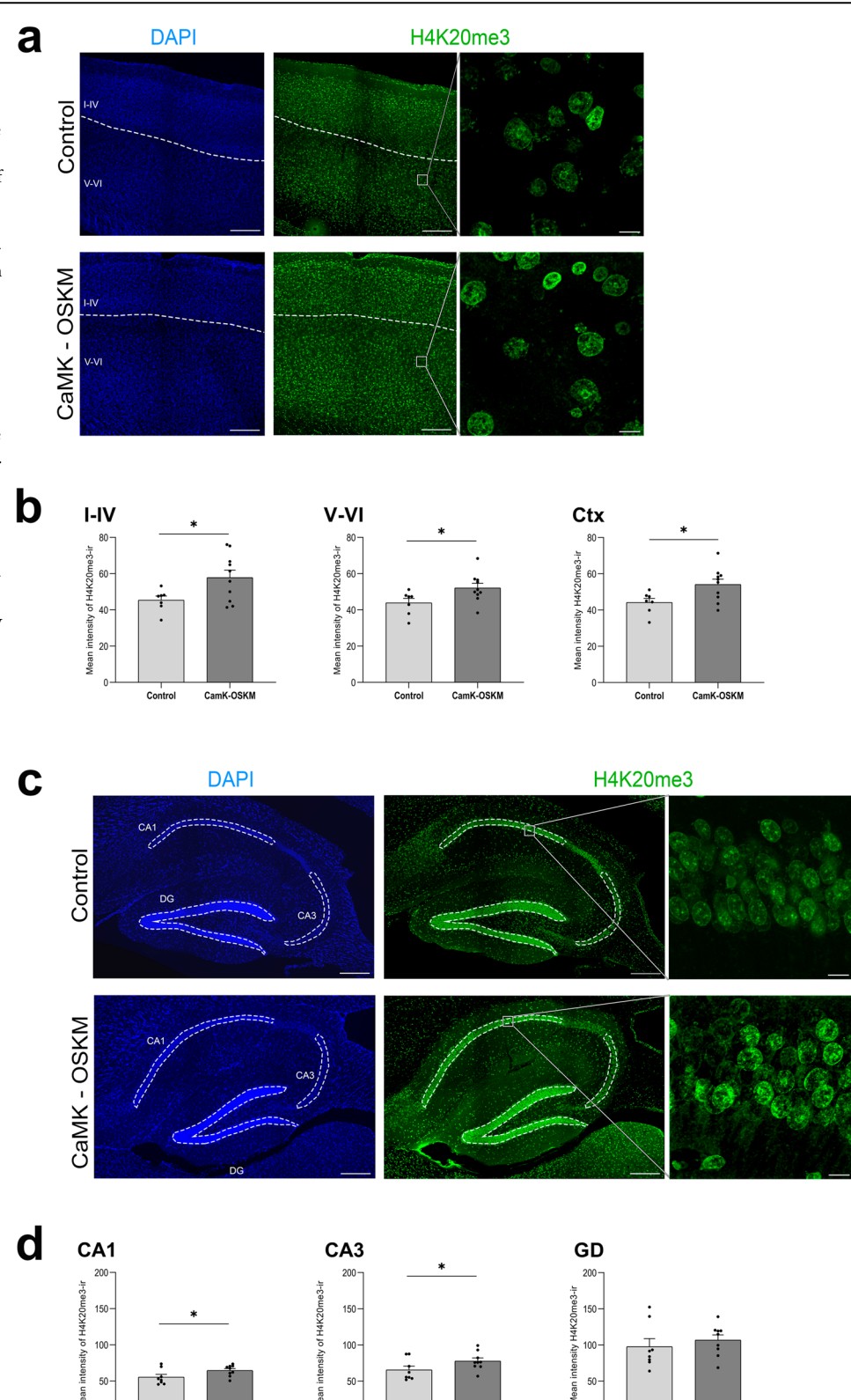

revealed data consistent with partial reprogramming processes, as groups of genes found differentially expressed, were related to cellular development of the nervous system or neuronal differentiation. In addition, these changes were significantly confined to the neuronal component of the cerebral cortex, with a considerable number of the altered genes being associated with dendritic and postsynaptic structures, as well as postsynaptic neuronal activity. Additionally, we observed alterations in genes associated with neuronal activity via receptors, along with changes in the expression of genes linked to behaviour or cognition. Taking all these changes into consideration, we aimed to determine whether neuronal partial reprogramming in these animals could result in alterations in the dynamics of memory-related circuits. Ultimately, these alterations could have contributed to the observed

cognitive improvement in these mice. For this purpose, we examined the expression of c-Fos, a protein transiently expressed during different memory tasks and whose inhibition has been found to impair memory formation[47]. We found a higher density of active neurons in the granular cell layer of DG and in the pyramidal cell layer of CA1 after performing memory tests in which these areas are involved, such as the recognition of new objects[48] or spatial memory[49]. Our results pointed out that partial reprogramming of neurons involved in memory processing, made them more active in memory circuits, improving their cognition functions.

Furthermore, the extracellular matrix has been considered a key regulator of neuronal activity, through the control of receptors and ion channels or spine plasticity[50]. It plays a crucial role in brain plasticity, a phenomenon that decreases with age[41]. For example, aggrecan is a key component of perineuronal networks in the adult brain, and its loss induces a permanent state of plasticity similar to the critical period in young mice, when cortical circuits are particularly malleable[41]. Similar to our initial observations in young mice, transcriptomic analysis of α-CaMKII-OSKM adult mice also revealed significant effects in terms of cellular adhesion and in extracellular matrix organisation due to partial reprogramming in neurons. We found that the overall downregulation of extracellular matrix genes was accompanied by a decrease in aggrecan levels throughout the cerebral cortex, particularly in deeper layers of the neocortex, where the YF expression was the highest. As aggrecan expression typically increases with age[42], the overexpression of YF appears to induce a more youthful state of the extracellular matrix.. Given that previous studies have shown an association between a decrease in extracellular matrix components and memory enhancement[51–53], it is possible that the reduced expression of aggrecan throughout the cerebral cortex, as observed in our results, could contribute to the memory improvement resulting from neuronal partial reprogramming.

Next, we aimed to determine whether the expression of Yamanaka factors in CamKIIα-neurons alone could dedifferentiate them, leading to earlier developmental stages or states of immaturity. In our adult transgenic α-CaMKII-OSKM animals with cyclic expression of YF, we did not observe immature neurons expressing doublecortin outside the dentate gyrus, which is normally the only region, along with the subventricular zone, in adult mammals where they can be found. This would confirm that expression of YF in cortical neurons, seems not to be sufficient to bring neurons to immature stage, something that had been previously described in neuronal cultures[17]. Furthermore, when focusing on the dentate gyrus, we did not observe differences in the density of these immature neurons compared to the control group, nor in the density of intermediate neural progenitors. This also confirms that neuronal YF expression does not significantly impact the generation of new neurons in the adult subgranular zone of the dentate gyrus (DG), as previously reported with ubiquitous YF expression[13]. We also did not observe an increase in proliferation, as studied with a thymidine analogue (CldU). The insufficient ability of ectopic expression of Yamanaka factors to revert mature neurons to an embryo-like stage[17], could explain the lack of significant differences in proliferation observed in adult mice during YF induction.

Finally, given the upregulation of genes related with histone methyltransferases like KTMT2D, we decided to study specific age-related epigenetic markers. There is growing evidence that manipulating the levels of some epigenetic markers can extend the lifespan by increasing genome stability[54], and these ones can apparently be slowed and even reversed by reprogramming[55]. H4K20me3 is a marker of heterochromatin and transcriptional repression, which decreases in the brain of the accelerated senescence-prone mouse mode at 12 months of age[56]. Our data have shown that the partial reprogramming of neurons led to a generalised increase in the methylation of histone H4 Lysine 20 marker throughout the neocortex and, also, in CA3 and CA1 pyramidal cell layers. These results suggest an epigenetic rejuvenation of neurons through partial reprogramming, reversing the decline of this aging-associated marker through epigenetic re-methylation. The increase in H4K20me3 methylation may be linked to the significant upregulation of the Zfp386 gene transcription observed in this

study. This protein associates with KAP1 and DNA methyltransfrases like DNMTs[57] which have been found to act on epigenetic marks such as H4K20me3, in order to silence LINE-1 elements[58]. It is also important to note that the increase of these retrotransposons have been correlated with aging[59].

In summary, we observed a strong correlation between cyclic expression of Yamanaka factors in neurons and age-dependent cellular and molecular changes accompanied by an improvement in mouse performance in the object recognition test. The in vivo partial reprogramming in adult mice of neurons has led to a reverses age-associated phenotypes, characterized by increased neuronal activity associated with memory circuits, by the reorganization of the extracellular matrix into more permissive stages for neuronal plasticity and by the hypermethylation of epigenetic markers like H4K20me3. All of these changes, resulting from cyclic rather than continuous YF expression, can have potentially contributed to memory enhancement, leading to the conclusion that partial reprogramming in a subpopulation of neurons is sufficient to bring about cognitive improvements in mice. Further analyses should lead to new strategies for expanding the regenerative capacity of brain tissue to try to prevent age-associated neurodegenerative diseases. The advantage of this strategy, which exclusively restricts partial reprogramming to neurons, would lie in the post-mitotic nature of neurons. This characteristic helps avoid potential collateral effects associated with Yamanaka factors, such as dedifferentiation, which can lead to teratoma formation.

## Methods

### Animals
A total of 49 male and female mice (C57BL/6 genetic background) were bred in the animal facility at the Centro de Biología Molecular Severo Ochoa, housed in a specific pathogen-free colony facility under standard laboratory conditions following the European Community Guidelines (Directive 86/609/EEC), and handled in accordance with European and local animal care protocols. Animal experiments received the approval of the CBMSO's (AECC-CBMSO-13/172) and national (PROEX 102.0/21) Ethics Committees. They were housed 4–5 per cage with food and water available *ad libitum*, and maintained in a temperature-controlled environment on a 12/12-h light/dark cycle with light onset at 8 a.m.

### Transgenic mice
In order to assess partial reprogramming restricted to neurons, a new murine model (double-transgenic) was designed crossing two different transgenic lines (Fig. 1a): 1) A transgenic mouse line, carrying the transcriptional transactivator (tTA). The tTa transgene is under the control of the Calcium Calmodulin Kinase II alpha promotor[60]. This tTa line allows restricted and constitutional expression in the CNS, with high expression in the forebrain[60,61], in areas related with spatial learning and memory[62]. 2) A mouse line which had a single copy of a doxycycline-inducible polycistronic lentiviral cassette encoding the four murine factors OSKM associated with a TetO promoter (Fig. 1b)[10]. Mouse offspring that only inherited this transgene, without the transactivator needed to express OSKM, were considered in this work as our control.

### In vivo induction of Yamanaka factors (YF)
In this new transgenic mouse line, the expression of Yamanaka factors (YF) is inhibited by administering doxycycline (2 mg/mL) in drinking water (Tet-off system). Consequently, to induce cellular partial reprogramming through YF expression, we had to discontinue doxycycline administration. The continuous expression protocol involved the continuous administration of water without doxycycline, while the cyclic protocol consisted of water administration for 3 days per week and doxycycline administration on the remaining 4 days.

The animals were divided into three experimental conditions: 15 mice, 3 months old, received continuous water administration (without doxycycline) from birth; 16 mice, received continuous water administration (without doxycycline) only from 6 months old until 11 months old; and 18

mice, received cyclical weekly water administration, with and without doxycycline, also starting from 6 months old until 11 months old (Fig. 1c). The control group for each experiment consisted of mice lacking the transcriptional transactivator (tTA). It is essential to highlight that the mice in the control group of each experiment received the same doses of doxycycline as their double-transgenic siblings, which was different between three experiments. In all cases, mice were perfused while the transgene induction process was active, meaning there was no doxycycline in the drinking water.

### CldU injection

The mice received a total of three injections, spaced every 2 h (200 μl/injection, intraperitoneally in alternative sites of the body) of 5-Chloro-2'-deoxy-Uridine (CldU, Sigma reference C6891; i.p. 42.75 mg/Kg body weight) three weeks before the sacrifice. This approach allowed us to monitor the effects of the treatments on the proliferation of new cells in the subgranular zone (SGZ), a key region of the brain for neurogenesis, labelling three-week-old new generated cells.

### Behavioural testing

Both Open Field (OF) and Novel Object Recognition (NOR) tests were performed as described previously[13]. Locomotor activity as well as anxiety and depression-like behaviours in old mice were evaluated using OF. This test was used as habituation of NOR test. In brief, on the first day, the mice were placed individually for 10 min in a 45×45-cm isolated plastic box. Each session was recorded and analysed using AnyMaze software (Stoelting Co., Wood Dale, IL, USA). On the second day, mice were placed in the same box for 5 min, allowing them to explore two identical objects: A and B (two black rooks). Both objects were placed on the long axis of the cage, each 13 cm from the cage end. After each exposure, the objects and the cage were wiped with 70% ethanol to eliminate odours which potentially could condition successive mice behaviour. Two hours after the familiarisation trial, each mouse was released into the box with the same object previously used (object A) and a new one (object C, a tower of coloured plastic pieces), instead of the object B (Short-term memory test). The position of object C was the same where object B was in the familiarisation trial. The mice were given 5 min to explore the box. Animals were considered to show recognition when their heads were less than 3 cm from the object.

The memory index (MI) was used to measure recognition memory performance. It was defined as the ratio of the times the mouse made contact (entries) with the new object (C) to the times it had made contact with both objects (A + C) (MI = [C/(A + C)]·100). AnyMaze software was used to calculate the number of contacts with objects by mice.

To assess spatial memory, an Y maze test based on published protocols with modifications[63] was performed. This test is based on the innate curiosity of rodents to explore novel areas. Firstly, mice were placed into one of the arms of a Y-maze black apparatus, which is made up of three plastic arms forming a "Y" shape. Mice were then allowed to explore the maze for 10 minutes, with one of the arms closed (training trial). After one-hour interval, mice were returned to the Y maze into the same arm (start arm) and were allowed to explore all three arms of the maze for 5 minutes. The number of entries in each arm were registered from video recordings, and analysed by AnyMaze software. The memory index (MI) was used to measure recognition memory performance. It was defined as the ratio of entries exploring the arm which was closed to the entries exploring both arms, the closed and opened ones.

It is important to note that all behavioural tests were always conducted on days when the inducer system for the expression of Yamanaka factors was active, meaning the mice did not have doxycycline in their drinking water.

### Sacrifice and tissue processing

Similar to a previous study in which the effects of ubiquitous YF expression on the hippocampus were analysed[13], the mice were sacrificed at 11 months of age, a point at which they are considered middle-aged adults. They were anaesthetised with an intraperitoneal pentobarbital injection (Dolethal, 60 mg/kg body weight) and transcardially perfused with saline. Brains were separated into two hemispheres. One hemisphere was removed and fixed in 4% paraformaldehyde in 0.1 M phosphate buffer (PB; pH 7.4) overnight at 4 °C. The next day, it was washed three times with 0.1 M PB and cut along the sagittal plane using a vibratome (Leica VT2100S). Serial parasagittal sections (50 mm thick) were cryoprotected in 30% sucrose solution in PB and stored in ethylene glycol/glycerol at -20 °C until they were analysed. The hippocampus, the neocortex and cerebellum of the other hemisphere were rapidly dissected on ice and frozen in liquid nitrogen for further analysis.

### Quantitative RT-PCR (qPCR)

Gene expression was determined by qPCR. RNA was isolated with the kit Maxwell 16 miRNA Tissue (AS1470, Promega) and the reverse transcription (RT) was performed using the iScript cDNA Synthesis Kit (1708891, Bio-Rad), according to the manufacturer's protocols in both cases. qPCR was performed using Fast EvaGreen Supermix (Biorad, CN 172-5204) and using CFX384 Real Time System C1000 Thermal Cycler (Bio-Rad) equipment.

The reaction per well is 10 μL and contains: 0.5 μL of cDNA template per sample to bring it to 10 ng/well and bringing this volume to 4 μL with $H_2O$, in addition to 5 μL of the EvaGreen Supermix and 1 μL per oligonucleotide pair at 5 μM. The primers used, targeted to E2A-cMyc to assure the expression of the full OSKM transgene (Fig. 1a), are as follow: E2A-cMyc forward primer: 5′-GGCTGGAGATGTTGAGAGCAA-3′, E2A reverse primer: 5′-AAAGGAAATCCAGTGGCGC-3′; BACT forward primer: 5'-CTAAGGCCAACCGTGAAAAG-3'; BACT reverse primer: 5'-ACCAGAGGCATACAGGGACA-3'; 18 S forward primer: 5'-CTCACCACGGGAAACCTCAC-3'; 18 S reverse primer: 5'-CGCTC CACCACCTAAGAACG-3'. qPCR amplification of genes was performed for 40 cycles of 95 °C for 5 s and 60 °C for 5 s. The BACT and 18 S genes were used as normalised genes. No amplification from the no-template control (NTC) was observed for genes of interest. Each primer pair showed a single, sharp peak, thereby indicating that the primers amplified only 1 specific PCR product. Three technical replicates per gene were used.

### RNA-seq analysis

RNA was isolated in Maxwell 16 Instrument using a Maxwell 16 LEV simply RNA tissue kit (Promega, P.N. AS1470), according to the manufacturer's instructions (Promega, P.N. AS1470). The sample concentration, purity, and integrity of RNA were quantified using a Nanodrop One spectrophotometer (Thermo Fisher Scientific). The RNA Integrity of the samples was checked using an Agilent 2100 Bioanalyzer.

For RNA-seq, the original data included 1 × 13 read sets in FASTQ format, sequenced in two runs, and quality analyses were performed over reads using FastQC1 software. The reference genome and annotation file of *M. musculus* (mm10) were downloaded from UCSC ftp site2 and GENE-CODE3, respectively. The reads were aligned against the *M. musculus* genome using Hisat24 aligner, a fast and sensitive alignment program for mapping next-generation sequencing reads to a reference genome.

Htseq-count7 was used to count the reads mapped to each feature. In the RNA-seq experiment, these features correspond to genes, where each gene was considered to be the union of all exons. The "intersection-strict" resolution mode was used, where reads were counted only if they were inside a gene or inside the exons of a gene. If a read was located in more than one gene, only the first read was considered. Differential expression analysis was performed using a R software package[64]. This software calculates gene and transcript expression levels under more than one condition and tests for significant differences using the negative binomial distribution, which estimates variance-mean dependence in RNA-seq count data. It is more sensitive and precise than other methods and maintains control over false-positive rates. Heatmap and hierarchical clustering of differentially expressed genes was performed using the heatmap function in stats package in R. For data analysis, we received support from the Genomics and NGS

Core Facility at the Centro de Biología Molecular Severo Ochoa (CBMSO, CSIC-UAM), which is part of the CEI UAM + CSIC, Madrid, Spain.

## Immunofluorescence

For immunofluorescence experiments, free-floating serial sections (50 μm) were first washed in 0,1 N PB three times and then preincubated for 2 h in PB with 0.25% Triton-X100 and 3% normal serum of the species in which the secondary antibodies were raised (Normal Goat or Donkey Serum). For a better signal of some primaries antibodies which are mainly present on the nucleus, free floating sections were pre-treated with 2 M HCl for 10 min at 35 °C. Subsequently, brain sections were incubated for 24 h at 4 °C in the same preincubation stock solution containing the primary antibodies in different combinations. After three washings in PB, the sections were incubated for 2 h at room temperature with the appropriate combinations of Alexa secondary antibodies with 1:500 concentration. Afterwards, three washes followed by incubation for 10 min in DAPI (1:5000, Merck, 268298), and then three more washes. Mounting was performed with FluorSave to preserve fluorescence in the stained tissues.

Primary antibodies used for immunofluorescence were as follow: rabbit Aggrecan (1:500, Merck, ab1031), rat CldU (1:200, Abcam, ab6326), rabbit c-Fos (1:500, Cell Signaling, #2250), guinea pig Doublecortin (1:250, Merck, ab2253), mouse H4K20me3 (1:200, Abcam, ab78517), goat Klf4 (1:250, R&D system, af3158), rabbit Tbr2 (1:200, Abcam, ab23345). Secondary antibodies were used as follows: donkey anti-goat Alexa-488 (Thermo Fisher, A-11055), goat anti-guinea pig Alexa-488 (Thermo Fisher, A-11073), donkey anti-mouse Alexa-488 (Thermo Fisher, A-21202), anti-streptavidin Alexa-488 (Thermo Fisher, S-32254), donkey anti-rabbit Alexa-594 (Thermo Fisher, A-21207), donkey anti-goat Alexa-647 (Thermo Fisher, A-21447), goat anti-rat Alexa-647 (Thermo Fisher, A-21247).

For imaging, the same range of z-slices was obtained for each brain slide in each experiment, obtaining the entire labelling present along the Z-axis of the section. A 1 μm interval step size was used for image acquisition through separate channels with a 20x lens on the Olympus Spinning Disk SpinSR10, except in the case of histone methylation marker analysis, where a 0,4 μm interval was used with a 40x lens. Confocal settings (laser intensity, gain, pinhole) were kept constant for all images of each combination of antibodies, which were captured in the same confocal session. Nikon A1R+ confocal microscope was used to take some microphotographs for representative images and Adobe Photoshop (CS4) software was used to compose figures.

## Immunofluorescence quantifications

For each immunofluorescence quantification were used three brain slices from each animal. Sagittal sections between 0.60 and 1.20 mm (approximately) lateral to the midline in Paxinos and Franklin's Mouse Brain Atlas[65]. Using the DAPI channel, areas were previously selected as a ROI in order to measure the volume. All densities were quantified as cells/mm³. For the c-Fos immunoreactivity analysis, density was quantified dividing the number of c-Fos immunoreactive cells by the volume of the granular cell layer and pyramidal cell layer of CA1 and CA3 regions. In the analysis of perineuronal nets (PNNs), the density was quantified by dividing the number of Aggrecan-immunoreactive PNNs in each brain slice by the volume of the region in which the analysis was conducted. For the analysis of neurogenesis, it was also quantified the density of CldU, Dcx and Tbr2 immunoreactive cells in the dental gyrus distinguishing subgranular zone (SGZ) and granular cell layer (GCL). The Dcx-immunoreactive cells were classified by differentiating different stages of maturation of young neurons, distinguishing three different cell types: type I, type II and type III. Type I corresponds to neuroblasts, type II to immature neurons with poorly developed dendritic trees and type III to young neurons, whose dendritic trees would be comparable to those of a mature neuron. Since the cell density in adults was very low, type II and III cells were pooled into a single group. Quantification of the percentage area of immunoreactivity produced by aggrecan and Klf4 antibodies in the somatosensorial neocortex and in hippocampus was performed from the maximum Z-projection of immunofluorescent stain from each stack used. A constant predefined threshold was used for all experimental groups for each type of staining using Fiji software. Finally, for the analysis of H4K20me3 immunoreactivity, we previously selected the entire granule cell layer (GCL), the pyramidal stratum of CA3 and CA1, and a consistent area from the somatosensory cortex. An invariant subtract background and threshold was then set in Fiji for each stack projection. The area above the threshold was measured in Fiji and the mean fluorescence in the ROI was calculated and compared between the experimental groups.

## Western blot technique

Neocortical brain samples from CamK-OSKM and transgenic control mice were homogenized in 20 mM HEPES pH 7.4, 100 mM NaCl, 50 mM NaF, 5 mM EDTA, and 1% Triton X-100 supplemented with protease and phosphatase inhibitors. The protein concentration of each homogenate was determined by the Bradford method using the BCA test (Thermo Fisher, Waltham, MA, USA). Samples were separated on 10% SDS-PAGE and electrophoretically in the presence at 100 mV for approximately 1 h and then transferred to a nitrocellulose membrane (Schleicher & Schuell GmbH). Membranes were blocked by incubation with 5% semi-fat dried milk in PBS and 0.1% Tween 20 (PBS), and then stained with Ponceau dye (Ponceau 0.3% in TCA 3%) to check transfer efficiency. Membranes were incubated with the appropriate primary antibody (diluted in PBS) overnight at 4 °C. Rabbit anti-Aggrecan (1:5000, Merck, ab1031) and β-actin (1:10000; SIGMA A5441), as loading control, were used as primary antibodies. After three washes, the membrane was incubated with a horseradish peroxidase-anti-rabbit Ig conjugate (DAKO), followed by several washes in PBS-Tween 20. The membrane was then incubated for 1 min in Western Lightning reagents (PerkinElmer Life Sciences). Blots were quantified using the EPSON Perfection 1660 scanner and the Image J software plugin.

## Statistics and Reproducibility

All data are represented as mean ± SEM (Standard Error of the Mean) in a bar (one bar per column) plot, as well as the mean of each animal in a dot plot. Statistical analyses were performed with Prism 9 software (GraphPad). The Shapiro-Wilk test was performed on each group of data to test for distribution normality. An unpaired Student's t-test was performed, unless it was non-parametric, then the Mann-Whitney test was applied. One-way ANOVA (or Kruskal-Wallis when data were not parametric) was employed for more than two group comparison. A significance level of 0.05 was considered for all tests used in the study (*$p < 0.05$, **$p < 0.01$, ***$p < 0.001$, ****$p < 0.0001$). It was used the outlier calculator from GraphPad to identify and exclude significant outlier's values.

## Reporting summary

Further information on research design is available in the Nature Portfolio Reporting Summary linked to this article.

## Data availability

The raw data from RNA-seq in this study are available in European Nucleotide Archive. The raw data from RNA-seq data from transgenic animals that were not administered with doxycycline from birth are under the ENA accession number PRJEB56610. The raw data from transgenic animals with cyclic expression of Yamanaka Factors in adult age are under the ENA accession number PRJEB65922. The numerical data are available in Figshare at https://figshare.com/articles/dataset/In_vivo_cyclic_overexpression_of_Yamanaka_factors_restricted_to_neuronsreversesage-associated_phenotypes_and_enhances_memory_performance/25688931, as well as in Supplementary Data 1 and Supplementary Data 2 files.

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

## Acknowledgements

Work in the laboratory of FH is funded by grants from the Spanish Ministry of Economy and Competitiveness (Ministerio de Economía, Industria y Competitividad, Gobierno de España, PID2020-113204GB-I00). Work in the laboratory of JA is funded by grants from the Spanish Ministry of Economy and Competitiveness (PGC-2018-09177-B-100). The authors would like to thank Dr Manuel Serrano for the use of the TetO-OSKM mice and for his helpful comments about this work. We thank Rocio Peinado-Cahuchola for the maintenance of our colony of transgenic animals. We are also grateful to Raquel Cuadros for technical assistance. We thank Nuria de la Torre for editorial assistance. We thank the Microscopy and Genomic Facilities at CBMSO-CSIC for the immunofluorescence analysis of biological samples and RNA/DNA analysis. The next-generation sequencing (NGS) data analysis has been performed by the Genomics and NGS Core Facility (GENGS), at the Centro de Biología Molecular Severo Ochoa (CBMSO, CSIC-UAM) which is part of the CEI UAM + CSIC, Madrid, Spain - http://www.cbm.uam. es/genomica - we acknowledge the computer resources and assistance provided by Centro de Computación Científica - Universidad Autónoma de Madrid (CCC-UAM) https://www.ccc.uam.es/.

## Author contributions

AAF and FH designed the experiments. AAF and MRL carried out the experimental studies and analyses. LVS performed the qPCR experiments and supervised transcriptomic studies. AAF, MRL and FH wrote the manuscript. FH and JA conceptualized the project. All authors have reviewed the manuscript and have reached a consensus on the final published version.

## Competing interests

The authors declare no competing interests.

## Additional information

**Peer review information** : *Communications Biology* thanks the anonymous reviewers for their contribution to the peer review of this work. Primary Handling Editors: Christoph Anacker, George Inglis and Benjamin Bessieres. A peer review file is available.

