## [Peer review file · Communications Biology]

Reviewers' comments:

Reviewer #1 (Remarks to the Author):

In Vivo Cyclical Overexpression of Yamanaka Factors Restricted to Neurons Reverses Age-Associated Phenotypes and Enhances Memory Performance by Anton-Fernandez and Roldan-Lazaro et. al.

The study investigates the impact of in vivo cyclic overexpression of Yamanaka factors (YF) specifically within neurons, aiming to reverse age-associated phenotypes and enhance memory performance. The authors developed a new mouse model using an inducible CaMK-specific activation of YF in adult mice. Interestingly, mortality and hydrocephaly was observed under constant activation during development, but no mortality when activated in the adults with both protocols, continuous and cyclic, and improve in cognitive abilities was seen in the cyclic protocol. While the overall framework of the study is well-constructed, there are certain aspects that require clarification and further elaboration.

1- Controls and Genotypes:

One critical aspect that needs clarification is the genotype of the controls used throughout the study. A comprehensive specification of the control genotypes and the rationale behind their selection will enhance the reader's understanding and assist in interpreting the results. It is essential to differentiate and thoroughly describe controls used for cyclic and continuous inductions, as this may influence the observed effects. For example in 2B and 2F is that the control was much higher in the continuous than in the cyclic. Therefore, the changes in these can be due to the control used, and not a real improvement.

2- Protein Half-Life and Expression Dynamics:

There is a lack of information regarding the lifetime of the Yamanaka factors. Given the cyclical nature of the induction and the potential impact on protein expression dynamics, it would be insightful to explore the half-life of the Yamanaka factors. This information can contribute to understanding the duration and persistence of the induced effects.

3- Efficiency of Reprogramming:

The authors briefly touch upon the inefficiency of reprogramming, suggesting the need for a marker to approximate the number of cells undergoing changes. Since reprogramming is a very inefficient process, even when all the cells are infected, for example in the process of producing iPSC. The authors should find a marker to approximate the number of cells with the inducing changes, for example number of cells per field expressing c-fos compared to the number of cells expressing KLF4. This should give an idea on the extend of the effect of the YF on the genome, (expression of Yamanaka factors vs changes in cellular markers, for example density of c-fos compare to density of klf, or area occupied, to have an assessment for example, that if X numbers of cells express YF but only 1/100 expressed c-fos, or has changes in methylation, etc. This could give a better understanding of the process and efficiency of the method.

4- RNA Expression and Neuronal Dilution:

Given that not all neurons may express all Yamanaka factors, potential dilution effects in the RNA expression results need consideration. To address this, exploring co-localization of YF at the protein level could help validate and strengthen the RNA-based findings. This would enhance the reliability

of the results by confirming the presence of YF in the same cells where changes in RNA expression are observed. Could you add a figure showing in the RNA seq the levels of their expression?

5- Comparison Between Cyclical and Continuous Induction:

An important point for clarification is whether the controls used for cyclic and continuous inductions are comparable. Differences observed, such as higher control levels in continuous induction compared to cyclic, need thorough examination to ensure that changes are attributed to the induction strategy rather than differences in control conditions. Directly comparing cyclic and continuous inductions is crucial for drawing accurate conclusions.

6- Correlation and causation

The study presents promising findings, addressing the aforementioned points although the paper is mainly correlative, therefore, the correlative results should be toned down in the text through the paper.

Reviewer #2 (Remarks to the Author):

In this manuscript, Anton-Fernandez et al aim to study the effects that partial reprogramming in a subpopulation of neurons in the cerebral cortex exert on neurogenesis, cognitive performance and mature neurons dedifferentiation. Authors use two different methodological strategies to stimulate the expression of Yamanaka Factors (YF). The work is original and innovative, the model constitutes a new approach to induce the expression of YF in the adult brain and the description of the model is interesting. The manuscript has been carefully written and is easy to follow. However, some aspects of the manuscript could use some improvements in order to demonstrate the proposed hypothesis and some of the statements claimed in the discussion.

Major concerns are listed below:

1. In fig 1 and suppl fig 2, authors only test for the presence of KLF4. How do authors know that all YF are being expressed in the same regions? Are they all expressed equally throughout the different regions?
2. Authors aim to replicate the effect of YF expression in vitro previously found by Kim and colleagues 2011 using an in vivo model. They discuss that they are obtaining a more youthful phenotype without affecting proliferation. However, proliferation has not been tested by authors. Detection of proliferation markers are necessary to support the idea that an increased proliferation is not involved.
3. In suppl fig 4, authors show DCX+ cells in control and transgenic mice. However, representative images of the cells that are being quantified in each graph are not shown. Representative images are necessary.
4. Authors show that the continuous induction model leads to a larger expression of KLF4 compared to the cyclic model. However, the cyclic model induces larger alteration in behavioral phenotypes. How do authors explain this result?
5. In figure 6, authors test the expression of aggrecan by immunohistochemistry and quantify the area occupied by aggrecan. This would give an approximate idea of total aggrecan expression. However, authors need to quantify the number of cells that express aggrecan in each condition and

use additional techniques to test the total levels of aggrecan expression (i.e. western blot)

6. In figure 7, authors use immunohistochemistry to analyze the methylation of histone H4. However, additional methods to test the methylation levels of H4K20 would be more precise. Authors could use western blott and show the total expression of H4 and the percentage of the total protein that has been methylated in response to the treatments in each brain area for example.

Reviewers' comments:

Reviewer #1 (Remarks to the Author):

In Vivo Cyclical Overexpression of Yamanaka Factors Restricted to Neurons Reverses Age-Associated Phenotypes and Enhances Memory Performance by Anton-Fernandez and Roldan-Lazaro et. al.

The study investigates the impact of in vivo cyclic overexpression of Yamanaka factors (YF) specifically within neurons, aiming to reverse age-associated phenotypes and enhance memory performance. The authors developed a new mouse model using an inducible CaMK-specific activation of YF in adult mice. Interestingly, mortality and hydrocephaly was observed under constant activation during development, but no mortality when activated in the adults with both protocols, continuous and cyclic, and improve in cognitive abilities was seen in the cyclic protocol.

While the overall framework of the study is well-constructed, there are certain aspects that require clarification and further elaboration.

1- Controls and Genotypes:

One critical aspect that needs clarification is the genotype of the controls used throughout the study. A comprehensive specification of the control genotypes and the rationale behind their selection will enhance the reader's understanding and assist in interpreting the results. It is essential to differentiate and thoroughly describe controls used for cyclic and continuous inductions, as this may influence the observed effects. For example in 2B and 2F is that the control was much higher in the continuous than in the cyclic. Therefore, the changes in these can be due to the control used, and not a real improvement.

We appreciate the reviewer's observation as indeed it may not be clear in the text whether the mice used as controls were the same or not for the different experiments conducted. Transgenic controls from different experiments received different doses of doxycycline. For adult mice experiments, the control mice (along with their sibling double transgenics) received doxycycline for the first 6 months of life. At that point, one group ceased doxycycline intake entirely, while another group continued to take it cyclically (4 days a week). The transgenic controls used in both cyclical and continuous induction experiments were littermate siblings of the double transgenics from each respective experiment. Therefore, both control groups received different treatments of doxycycline.

To clarify this point, we have added the following sentences in the page 4 of manuscript:

Line 121: "Mouse offspring that only inherited this transgene, without the transactivator needed to express OSKM, were considered in this work as our control in each of the experiments of this work."

Line 136: “The control group for each experiment consisted of mice lacking the transcriptional transactivator (tTA). It is essential to highlight that the mice in the control group of each experiment received the same doses of doxycycline as their double-transgenic siblings.”

2- Protein Half-Life and Expression Dynamics:

There is a lack of information regarding the lifetime of the Yamanaka factors. Given the cyclical nature of the induction and the potential impact on protein expression dynamics, it would be insightful to explore the half-life of the Yamanaka factors. This information can contribute to understanding the duration and persistence of the induced effects.

The reviewer is right about the interest in understanding the half-life of expression of factors with this type of cyclic induction. Therefore, we have processed and analyzed the Klf4 Yamanaka factor protein expression in mice perfused at different times that followed the cyclical induction protocol during a single week session. All siblings studied were double transgenic α -CaMKII-OSKM mice.

A scheme of this protocol has been included in the supplementary information document, in a new supplementary figure (Suppl.Fig.2). The results have shown that indeed, after 3 days of induction, there is a significant increase in the expression of Klf4. This induction drops to day 0 levels after 4 days of continuous doxycycline administration.

We have added the following paragraph:

Line 432: The cyclical expression of Yamanaka factors was confirmed by the histological study of Klf4 protein expression at different time intervals during a week cyclical protocol (Suppl.Fig.2A-B). The analysis of immunofluorescence obtained from Klf4 Yamanaka factor expression have shown that indeed, after 3 days of induction, there is a significant increase (p -value= 0.0043) in the expression of Klf4 Yamanaka factors (Suppl.Fig.2C). This induction returns to day 0 levels after 4 days of continuous doxycycline administration (p -value=0.0146).

3- Efficiency of Reprogramming:

The authors briefly touch upon the inefficiency of reprogramming, suggesting the need for a marker to approximate the number of cells undergoing changes. Since reprogramming is a very inefficient process, even when all the cells are infected, for example in the process of producing iPSC. The authors should find a marker to approximate the number of cells with the inducing changes, for example number of cells per field expressing c-fos compared to the number of cells expressing KLF4. This should give an idea on the extend of the effect of the YF on the genome, (expression of Yamanaka factors vs changes in cellular markers, for example density of c-fos compare to density of klf, or area occupied, to have an assessment for example, that if X numbers of cells express YF but only 1/100 expressed c-fos, or has changes in methylation, etc. This could give a better understanding of the process and efficiency of the method.

Following the suggestion of the referee we have correlated data obtained from Klf4 immunoreactivity expression with c-fos+ cell density in the hippocampus from α -CaMKII-OSKM mice. We have added a new supplementary figure (Suppl.Fig.4) and this paragraph in page 14 of results:

Line 546: “Additionally, we have found in these mice a significant inverse correlation (R=0.9639) between levels of Klf4 expression and density of c-Fos-immunoreactive cells (Suppl.Fig.4). Excessive expression of Klf4 led to lower increase of c-Fos-immunoreactive cell density in the hippocampus of α -CaMKII-OSKM transgenic mice. These results are consistent with those found in α -CaMKII-OSKM mice with continuous induction of the YF. In these mice, where Klf4-YF expression is very high, we observed worse cognitive performance compared to those with cyclical induction. All these results underscore once again the importance of the level of induction of the YF. Moderate rather than excessive induction is what achieves beneficial effects on the cognition of aged mice.”

4- RNA Expression and Neuronal Dilution:

Given that not all neurons may express all Yamanaka factors, potential dilution effects in the RNA expression results need consideration. To address this, exploring co-localization of YF at the protein level could help validate and strengthen the RNA-based findings. This would enhance the reliability of the results by confirming the presence of YF in the same cells where changes in RNA expression are observed. Could you add a figure showing in the RNA seq the levels of their expression?

We agree with the reviewer regarding the limitations of the study concerning the potential masking of observed effects since only a population of neurons expresses the factors. Additionally, it is important to note that the transcriptomic analysis conducted in this study was performed using bulk RNA sequencing, so we cannot obtain data from specific individual cells; rather, we have data from the entire hippocampus, which may have potentially diluted the effects further. Therefore, we cannot confirm the presence of the factors in the same cells where changes in RNA expression are observed.

5- Comparison Between Cyclical and Continuous Induction:

An important point for clarification is whether the controls used for cyclic and continuous inductions are comparable. Differences observed, such as higher control levels in continuous induction compared to cyclic, need thorough examination to ensure that changes are attributed to the induction strategy rather than differences in control conditions. Directly comparing cyclic and continuous inductions is crucial for drawing accurate conclusions.

We agree with the reviewer on the importance of this point. In line with the aforementioned discussion, the controls used in the experiments of cyclical and continuous induction are not comparable, as they have both ingested significantly different amounts of the antibiotic doxycycline. With the changes made in the manuscript as described earlier, we believe it would be sufficient to clarify this point.

6- Correlation and causation

The study presents promising findings, addressing the aforementioned points although the paper is mainly correlative, therefore, the correlative results should be toned down in the text through the paper.

We do agree with the reviewer point of view. We have modified the following paragraph at the end of the discussion:

Line 766: In summary, we observed a strong correlation between cyclic expression of Yamanaka factors in neurons and age-dependent cellular and molecular changes accompanied by an improvement in mouse performance in the object recognition test. The in vivo partial reprogramming in adult mice of neurons has led to a reverses age-associated phenotypes, characterized by increased neuronal activity associated with memory circuits, by the reorganization of the extracellular matrix into more permissive stages for neuronal plasticity and by the hypermethylation of epigenetic markers like H4K20me3. All of these changes, resulting from cyclic rather than continuous YF expression, can have potentially contributed to memory enhancement, leading to the conclusion that partial reprogramming in a subpopulation of neurons is sufficient to bring about cognitive improvements in mice. Further analyses should lead to new strategies for expanding the regenerative capacity of brain tissue to try to prevent age-associated neurodegenerative diseases.

Reviewer #2 (Remarks to the Author):

In this manuscript, Anton-Fernandez et al aim to study the effects that partial reprogramming in a subpopulation of neurons in the cerebral cortex exert on neurogenesis, cognitive performance and mature neuron dedifferentiation. Authors use two different methodological strategies to stimulate the expression of Yamanaka Factors (YF). The work is original and innovative, the model constitutes a new approach to induce the expression of YF in the adult brain and the description of the model is interesting. The manuscript has been carefully written and is easy to follow. However, some aspects of the manuscript could use some improvements in order to demonstrate the proposed hypothesis and some of the statements claimed in the discussion.

Major concerns are listed below:

1. In fig 1 and suppl fig 2, authors only test for the presence of KLF4. How do authors know that all YF are being expressed in the same regions? Are they all expressed equally throughout the different regions?

Considering that *Klf4* expression is known to occur, due to the nature of the transgene, at least *oct4* and *sox2* would also be expressed. This is because *Klf4* is the third gene in the transgene construct, whose transcription occurs sequentially, meaning that to reach *klf4*, the other two genes must have been transcribed first. Thus, the expression pattern that we have described in different regions with KLF4 would coincide with that of the transgene (i.e., the other Yamanaka factors). Additionally, it should be noted that generally antibodies against these factors do not work well in tissue except for *klf4* or *sox2*. However, the latter would not be reliable since it also marks a significant population of astrocytes and progenitor cells in the hippocampus.

To make clearer this issue, we have added to the manuscript the following sentence in page 9:

Line 357: It is important to note that due to the nature of the transgene, the expression of *Klf4* (located in the third position within the gene order of the construct, after *Oct4* and *Sox2*) implies that the other genes should have also been expressed in the neuron, so the expression pattern observed with *Klf4* would correspond to that of the transgene as a whole.

2. Authors aim to replicate the effect of YF expression in vitro previously found by Kim and colleagues 2011 using an in vivo model. They discuss that they are obtaining a more youthful phenotype without affecting proliferation. However, proliferation has not been tested by authors. Detection of proliferation markers are necessary to support the idea that an increased proliferation is not involved.

Although there are several other methods (such as Ki67 or pH3) for the detection of cell division, the marker of choice in our MS is the thymidine analogue 5-Chloro-2'-deoxyuridine (CldU) that is incorporated into the DNA during the S-phase of the cell cycle. CldU is immunohistochemically detected and allows co-localization with other cell type markers. We believe that the results obtained with this thymidine analogue would be sufficient to confirm

alterations in proliferative levels within specific brain areas such as the dentate gyrus or cortex, where we did not observe changes with this marker (Supplementary Fig. 7). These results are commented in Results (Line 606-614) and Discussion (Line 741-747) sections

3. In suppl fig 4, authors show DCX+ cells in control and transgenic mice. However, representative images of the cells that are being quantified in each graph are not shown. Representative images are necessary.

New Supplementary Figure 7 has been updated with new representative images of different neurogenic markers (Suppl.Fig.7 A,B,C).

4. Authors show that the continuous induction model leads to a larger expression of KLF4 compared to the cyclic model. However, the cyclic model induces larger alteration in behavioral phenotypes. How do authors explain this result?

In the discussion, we previously noted that the reprogramming process can induce significant cellular stress. Therefore, if the expression of the factors in neurons is excessive or prolonged, it is likely to mask the improvements observed in partial (cyclic) reprogramming. This cellular stress could impact neuronal activity and circuit functionality, hindering the observation of memory processing enhancements. Furthermore, in this revised version of the manuscript, we have added, at the suggestion of the reviewers, data on the correlation between expression levels of the Yamanaka factors and c-Fos+ cells. We found that in mice expressing much higher levels of the Yamanaka factors, the increase in active cell density during memory tests was lower. This suggests that to observe cognitive improvement, moderate expression of the factors is crucial, inducing a rejuvenating effect on neurons without inducing cellular stress that could alter their functionality.

We have incorporated this analysis into the manuscript by adding a new supplementary figure and including the following paragraph on page 14 of the results section:

Line 546: "Additionally, we have found in these mice a significant inverse correlation ($R=0.9639$) between levels of Klf4 expression and density of c-Fos-immunoreactive cells (Suppl.Fig.4). Excessive expression of Klf4 led to lower increase of c-Fos-immunoreactive cell density in the hippocampus of α -CaMKII-OSKM transgenic mice. These results are consistent with those found in α -CaMKII-OSKM mice with continuous induction of the YF. In these mice, where Klf4-YF expression is continuous, we observed worse cognitive performance compared to those with cyclical induction. All these results underscore once again the importance of the level of induction of the YF. Moderate rather than excessive induction is what achieves beneficial effects on the cognition of aged mice."

5. In figure 6, authors test the expression of aggrecan by immunohistochemistry and quantify the area occupied by aggrecan. This would give an approximate idea of total aggrecan expression. However, authors need to quantify the number of cells that express aggrecan in each condition and use additional techniques to test the total levels of aggrecan expression (i.e. western blot)

We appreciate the suggestions provided by the reviewer as they have allowed us to strengthen the previously found results regarding the extracellular matrix. As proposed by the reviewer, we have analyzed the data of perineuronal net (PNN) units, obtaining, in line with the overall Aggrecan expression results, a significant decrease in PNN density in the neocortex but not in the hippocampus. These results have been added to Figure 6. Additionally, the general decrease in Aggrecan in the neocortex observed via immunofluorescence has been

corroborated by western blot analysis. This results have been illustrated in a new supplementary figure (Supplementary Figure 7). Additional sentences have been included:

Line 290: In the analysis of perineuronal nets (PNNs), the density was quantified by dividing the number of Aggrecan-immunoreactive PNNs in each brain slice by the volume of the region in which the analysis was conducted.

Line 311: Western blot technique is described

Line 574: In line with these results, the analysis of the density of perineuronal net units also shows a significant reduction following the induction of partial reprogramming in α -CaMKII-OSKM mice (Figure 6C). In contrast to the neocortical areas, noteworthy statistical differences were not found in the hippocampus between both experimental groups for the area occupied by the immunoreactive aggrecan matrix (Figure 6E), nor regarding the density of PNN units (Figure 6F). This general reduction found by immunofluorescence detection in aggrecan-immunoreactive extracellular matrix has been corroborated by Western blot technique (Suppl.Fig.6). We observed a highly significant decrease ($P=0.001$) in its expression following the cyclical induction of the Yamanaka factors. These data confirm the significant role of extracellular matrix reorganization during cyclical reprogramming processes in the brain.

6. In figure 7, authors use immunohistochemistry to analyze the methylation of histone H4. However, additional methods to test the methylation levels of H4K20 would be more precise. Authors could use western blott and show the total expression of H4 and the percentage of the total protein that has been methylated in response to the treatments in each brain area for example.

We again appreciate the suggestion made here by the reviewer. However, considering that the tissue samples we have used contain nuclei not only from neurons but also from all types of brain cells, finding differences in histone methylation seems very complicated, especially given that the changes would occur only in neurons. Therefore, we think that the immunofluorescence technique is the most accurate way to examine neuronal histone methylation changes in this case.

REVIEWERS' COMMENTS:

Reviewer #1 (Remarks to the Author):

No further comments.

Reviewer #2 (Remarks to the Author):

Having carefully reviewed the revised manuscript, I believe that authors have responded satisfactorily to all the comments and concerns raised. The attention to detail and the thoughtful revisions made have significantly improved the clarity of the work.